# Labits: Layered Bidirectional Time Surfaces Representation for Event Camera-based Continuous Dense Trajectory Estimation

## Abstract

Event cameras provide a compelling alternative to traditional frame-based sensors, capturing dynamic scenes with high temporal resolution and low latency. Moving objects trigger events with precise timestamps along their trajectory, enabling smooth continuous-time estimation. However, few works have attempted to optimize the information loss during event representation construction, imposing a ceiling on this task. Fully exploiting event cameras requires representations that simultaneously preserve fine-grained temporal information, stable and characteristic 2D visual features, and temporally consistent information density—an unmet challenge in existing representations. We introduce Labits: Layered Bidirectional Time Surfaces, a simple yet elegant representation designed to retain all these features. Additionally, we propose a dedicated module for extracting active pixel local optical flow (APLOF), significantly boosting the performance. Our approach achieves an impressive 49% reduction in trajectory end-point error (TEPE) compared to the previous state-of-the-art on the MultiFlow dataset. The code will be released upon acceptance.

## 1 Introduction

As an emerging visual modality, event cameras offer unique and practical advantages. Compared to conventional frame-based cameras, they provide higher temporal resolution, greater dynamic range, higher efficiency, and lower latency (Gallego et al. (2020)). Furthermore, under stable lighting, event cameras are primarily sensitive to the edges of moving objects, naturally filtering out stationary objects while tracking moving ones. Their ultra-high temporal resolution also enables smoother and more continuous target tracking. In recent years, numerous papers leveraging this feature of event cameras have addressed topics such as feature tracking (Messikommer et al. (2023)), optical flow generation (Wan et al. (2024)), and video interpolation (He et al. (2022)) based on events.

From an event camera's perspective, each moving point generates a discrete trajectory in the xyt space, with each triggered event representing a sampled point on this trajectory, along with its timestamp. The instantaneous velocity at any point on the trajectory can be calculated from the relative positions and times of these events using straightforward mathematical formula $v = \Delta x / \Delta t$, aligning with intuitive understanding. While real-world factors like rotations, depth movements, and multiple moving objects complicate pixel-wise speed, the trajectory information persists embedded within the event streams. Our target is to unearth these hidden treasures.

Currently, there are two main approaches to utilizing events. One is to directly construct events into a graph in the xyt space and input it into a GNN (Graph Neural Network) (Li et al. (2021)) or treat each event as a spike to be input into an SNN (Spiking Neural Network) (Kosta & Roy (2023)). The other is to first convert events into a dense representation and then input them into an Artificial Neural Network (ANN). Although GNNs appear efficient, constructing the graph is computationally and memory-intensive with increasing events, and GNNs suffer from over-smoothing as network depth increases (Chen et al. (2020)). Additionally, the graph conversion discards fine-grained spatial and temporal information, making position-sensitive scenarios problematic. These limitations restrict GNN applications in event-based vision, especially for low-level tasks. On the other hand, while SNNs offer high energy efficiency, they are more difficult to train and are less robust (Lee et al.

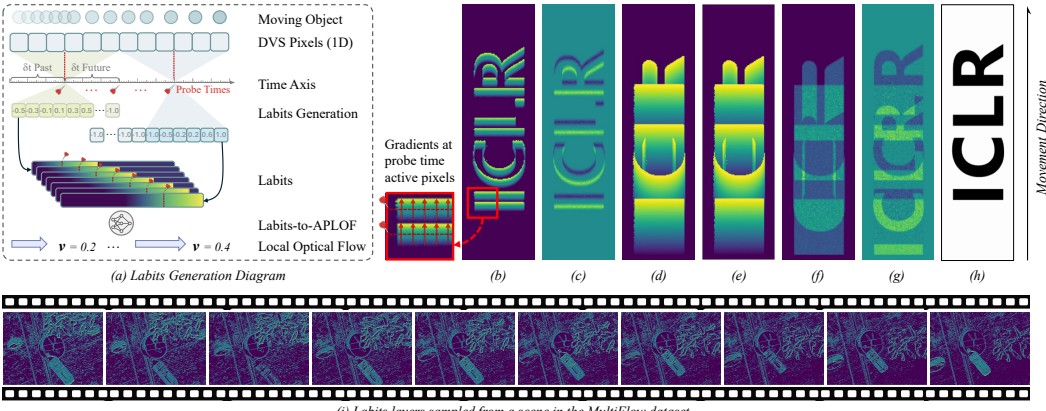

(a) Labits Generation Diagram  (b)  (c)  (d)  (e)  (f)  (g)  (h)

(i) Labits layers sampled from a scene in the MultiFlow dataset.

Figure 1: (a) Labits generation schematic: For a 1D event camera, at each pixel and probe time $pt_i$, the algorithm searches for the most recent past event within $\delta t$. If none is found, it searches for the next future event within $\delta t$. The Labits value is the normalized time difference between probe time $pt_i$ and the found event's timestamp, or -1 if no event is found. Labits can be converted to APLOF via a small model. (b)-(g): Single channel visualization of the following event representations (more details can be found in related works): (b) Labits (c) Voxel Grid (d) TORE Volume (e) Time Surface (f) Event Count (g) Event Frame (h) RGB frame of the moving target (i) Labits layers samples. Note that the first layer of TORE is exactly the same as time surface.

(2016)). Compared to deep learning, the SNN ecosystem is still in its early stages, and the maturity of neuromorphic hardware is insufficient to support large-scale, high-precision models in practical applications (Nunes et al. (2022)). These factors limit the widespread use of SNNs.

The mainstream approach in event-based vision remains converting the event stream into a dense representation before inputting it into an ANN for inference (Ye et al. (2023), wan2024event). However, existing representations have limitations that prevent event cameras from fully leveraging their strengths for dense trajectory estimation. Common representations like event frames (Rebecq et al. (2017)) and event counts (Maqueda et al. (2018); Zhu et al. (2018b)) entirely remove the temporal dimension by projecting all events onto a 2D plane. Event frames sum the polarities at each pixel, while event counts simply tally the events. Similarly, voxel grid (Zhu et al. (2019)) applies temporal quantization discards fine-grained temporal information, projecting events onto the nearest temporal grid using a bilinear sampling kernel. Although representations like time surfaces (Lagorce et al. (2016)) and TORE volumes (Baldwin et al. (2020)) preserve some temporal information, each pixel retains only the timestamps of the most recent events, ignoring earlier ones triggered by different objects. This leads to temporal occlusion and limits the ability to extract continuous, dense, and temporally consistent local motion information in the spatio-temporal ($xyt$) domain.

Dense trajectory estimation requires pixel-level movement sensitivity, stable and sharp 2D visual features, and consistent information density throughout the tracking duration. The final requirement is to ensure the representation does not excessively focus on events near the end of the time period while neglecting earlier occurrences. To fully exploit the potential of events for dense trajectory estimation, a representation that preserves all these characteristics is essential. In this paper, we propose Labits: **LA**yered **BI**directional **T**ime **S**urfaces, as a simple and elegant solution to these challenges. The impact of an effective event representation is significant. By simply switching to Labits, it can yield a **13%** improvement on TEPE, while incorporating an additional APLOF extractor can provide an extra **30%** reduction in error for dense trajectory estimation.

Our main contributions can be summarized as follows: 1. We proposed Labits, a novel synchronous event representation that is aware of event camera's asynchronous characteristic and keeps rich local movement trend information. 2. We trained a corresponding active pixel local optical flow estimator based on Labits layers, which utlizes intermediate motion information. 3. We achieved the SOTA performance on event-based dense trajectory estimation task, and the TEPE of our method decreased **49%** compared to the previous SOTA.

Table 1: A comparison of event representation methods used in deep learning, where $H$ and $W$ denote the height and width of the representations, respectively. This table extends the one presented in (Baldwin et al. (2022)) with newly included methods.

| Event Representation | Dimensions | Description | Characteristics |
|---|---|---|---|
| Event Frame | H, W | Image of event polarities | Discards temporal & polarity information |
| Event Count | 2, H, W | Image of event counts (EC) | Discards temporal information |
| Time Surface | 2, H, W | Image of most recent timestamp | Discards all prior time stamps |
| Averaged Time Surfaces | 2, H, W | Image of average timestamp for window | Discards temporal information |
| Inceptive Time Surfaces | 3, H, W | Image of filtered timestamps & EC | Discards temporal information |
| Event Spike Tensor | 2, B, H, W | 4D grid of convolutions | Temporally quantizes information into B bins |
| TORE Volumes | 2, K, H, W | 4D grid of last K timestamps | Discards timestamps prior to last K events |
| Voxel Grid | B, H, W | Voxel grid summing event polarities | Discards polarity information |
| Labits | B, H, W | Layered bidirectional time surfaces | Provides multi-layer local speed hints |

## 2 RELATED WORKS

**Event Camera:** Event cameras contain a bio-inspired dynamic vision sensor, where each pixel unit works asynchronously and triggers an event instantly when it detects a log-intensity change over a predefined threshold. Each event $\mathbf{e} = (t, x, y, p)$ records the spatial coordinate $(x, y)$ of the corresponding pixel position on the image sensor plane, the microsecond-level shooting timestamp $t$, and a binary polarity value $p$ that indicates the direction of brightness change (Zhang et al. (2024)). Event cameras are widely used in various computer vision and robotics tasks. They excel in motion-centric tasks like optical flow estimation and object or human pose tracking, as demonstrated in numerous studies (Hu et al. (2022); Wu et al. (2024); Chamorro Hernández et al. (2020); Zhang et al. (2023)). Their high temporal resolution and event-driven operation also enable innovative video processing techniques, including frame interpolation (Tulyakov et al. (2022); Sun et al. (2023); Liu et al. (2024)) and motion deblurring (Chen et al. (2024); Yang et al. (2024); Kim et al. (2024)). These applications leverage the unique characteristics of event cameras for detailed, dynamic scene analysis without the high data demands of traditional high-speed video recording.

**Event Representations:** In event-based vision, mainstream methods convert the event stream into a dense representation, then pass it to ANNs for various tasks (Bardow et al. (2016)). However, existing representations often fail to fully leverage the unique capabilities of event cameras.

Early representations attempt to convert event streams into intensity frames, highlighting moving edges and mimicking 2D features of traditional cameras. The event count representation (Maqueda et al. (2018); Zhu et al. (2018b)) sums the number of events per pixel within a time window, while the event frame (Rebecq et al. (2017)) sums event polarities. Both discard temporal information, obscuring events over time. The voxel grid representation (Zhu et al. (2019)) quantizes time and maps events to temporal grids using bilinear sampling. While better than event count, time information retention is still limited, and temporal obscuring persists.

Time surface-style representations form another key branch. The original time surface, or Surface of Active Events (SAE, Benosman et al. (2013)), encodes only the most recent event's timestamp per polarity at each pixel, disregarding prior events, no matter the scene's complexity. This leads to poor 2D pattern capture and temporal occlusion, where newer events overwrite earlier ones. Variants like the averaged time surface (Sironi et al. (2018)) reduce noise and mitigate occlusion, but reintroduce temporal obscuring and ambiguity. TORE volume (Baldwin et al. (2022)) preserves the most recent K events per pixel, creating multi-layer time surfaces. However, redundancy arises when the same object repeatedly triggers recent events at the same pixel, perpetuating temporal occlusion. This is partly due to the complex textures of real-world objects, which introduce numerous small edges that trigger events. All aforementioned representations, and others, are summarized in Table 2.

**Trajectory Estimation:** Continuous-time trajectory estimation was initially proposed for rolling shutter compensation Kerl et al. (2015). More closely related to our work is the regression of pixel trajectories, aligned with high-speed feature tracking in event cameras Gehrig et al. (2020); Alzugaray & Chli (2020). This work connects to methodologies described in Gehrig et al. (2024), where features are continuously tracked via the integration of Bézier curves, correlation map sequences, and image data. However, unlike our approach, their solution emphasizes visual pattern-based correlation while neglecting the fine-grained temporal information inherent in raw events, often leading to erroneous trajectory estimations.

## 3 Labits: Layered Bidirectional Time Surface

Event cameras have distinct advantages over traditional frame-based cameras due to their asynchronous nature. This characteristic enables event cameras to provide highly precise timestamps at the microsecond level, making them ideal for capturing instantaneous motion velocity. However, existing event representations fail to fully exploit these advantages. Most approaches convert the sparse and discrete event streams into frame-like structures that highlight 2D visual features such as moving edges and patterns, thus making the event modality compatible with conventional computer vision models. The transformation essentially forces event cameras to conform, fitting themselves into the framework dominated by frame-based cameras, rather than fully leveraging their own strengths and showcasing the unique advantages that conventional cameras cannot replicate.

Building event representations inevitably involves information loss. It's almost a trilemma to faithfully preserve the original fine-grained timestamp information, compile meaningful 2D visual patterns, and maintain historical movement information at intermediate times simultaneously. However, all three aspects are essential for accurate, dense, continuous-time trajectory estimation. Fine-grained event-level timestamp information is the unique strength of the event modality in movement prediction. 2D visual patterns form the basis for correlation-based tracking mechanisms, while intermediate movement information ensures stable estimation accuracy throughout the entire trajectory.

Therefore, we designed Labits: Layered Bidirectional Time Surfaces to meet these demands. The logic behind this representation is straightforward: a time-surface-like structure is essential for retaining the microsecond-level timestamp features. The accumulation time range of events should be strictly controlled to avoid issues with visual feature blurriness, so splitting the target time range into a series of smaller time bins is essential. Stopping at this stage creates a layered time surface.

However, under this scheme, only the last events' information is utilized within each time bin at each pixel, which is unpreferable. Moreover, when estimating the local speed at each time bin intersection, only past movement information is considered, so the prediction here is based on backward difference, where the error estimation is known to be $O(\delta t)$. Further reducing this error estimation is nothing difficult: simply consider future event if no event is observed in the past $\delta t$ search range. This change in representation is simple yet powerful: First, pixels ahead of the moving edges' direction often don't have triggered events in the near past, by incorporating future events, these originally empty pixels are assigned a meaningful value, thereby increasing the information density of the resulting representation. Second, the first and last event within each time bin at each pixel are both fully utilized. Especially when the time range is small, this sampling strategy is highly representative. Third, considering near past and future events simultaneously changes the basic representation unit from time bin to intermediate probe times. Local speed estimations at these probe times use central difference, reducing the estimation error to $O(\delta t^2)$ (detailed in the Supplementary Material).

Let $E = \{(t_n, x_n, y_n, p_n)\}_{n=1}^N$ represent the event stream, where $t_n$ denotes the timestamp, $(x_n, y_n)$ are the spatial coordinates, and $p_n$ is the polarity of the $n$-th event. The variables $\tau_{\text{start}}$ and $\tau_{\text{end}}$ refer to the first and last event timestamps, respectively, and $\tau_{\text{total}}$ is the total time duration of the events. The time interval between two adjacent probe times is denoted as $\tau_{\text{range}}$, and $\tau_i$ denotes the $i$-th probe timestamp. The output tensor $L \in \mathbb{R}^{B \times H \times W}$ is the Labits representation, where $B$ is the number of probe time points, $H$ and $W$ is the sensor height and width. The Labits generation algorithm is elaborated in Algorithm 1.

As shown in Algorithm 1, the backward and forward temporal search range at each probe time point is $t_{range}$, which corresponds to the time interval of each time bin. Under this scheme, the timestamp of the earliest and latest events within each time bin are recorded by the Labits layers at the two adjacent probe points surrounding the bin, further minimizing information loss.

Moreover, the design of Labits ensures that it maintains stability and consistency. The values in the each layer of Labits always lie within a fixed range, making Labits inherently robust against outliers, such as hot pixel noise, which can otherwise distort the normalization scale in other representations.

With Labits, dense instantaneous optical flows at active pixels can be generated across evenly-spaced probe times using straightforward neural networks, due to the clear correlation between Labits layer values and their corresponding local speeds. The resulting layered instantaneous optical flows could serve as a basis for future research into tasks requiring high temporal resolution and advanced motion understanding, such as event-based object or human pose tracking.

---

**Algorithm 1:** Labits Representation Generation

---

**Input** : $E = \{(t_n, x_n, y_n, p_n)\}_{n=1}^{N}$, where $t_n$ is timestamp, $(x_n, y_n)$ are spatial coordinates, and $p_n$ is the polarity of the event. Events are ordered by $t_n$ in ascending order.

**Output:** $L \in \mathbb{R}^{B \times H \times W}$, a tensor representing Labits. $B$ is the number of intermediate probe time points, $H$ and $W$ are the sensor height and width.

1 Set $\tau_{start} = t_1$ and $\tau_{\text{end}} = t_N$ (first and last event timestamps);
2 Compute total duration: $\tau_{\text{total}} = \tau_{\text{end}} - \tau_{\text{start}}$;
3 Divide $\tau_{\text{total}}$ into $B + 1$ equal intervals:

$$\tau_{\text{range}} \leftarrow \tau_{\text{total}}/(B+1), \quad \tau_i \leftarrow \tau_{\text{start}} + i \cdot \tau_{\text{range}}, \quad i \in \{1, \ldots, B\}$$

Initialize $L \in \mathbb{R}^{B \times H \times W}$ with $-1$;
4 **for** $i = 1$ *to* $B$ **do**
5    $E_{\text{prev}} \leftarrow \{e_n \mid \tau_i - \tau_{\text{range}} \leq t_n \leq \tau_i\}$; $E_{\text{future}} \leftarrow \{e_n \mid \tau_i < t_n \leq \tau_i + \tau_{\text{range}}\}$;
6    Define $T_{\text{prev}} \in \mathbb{R}^{H \times W}$ and $T_{\text{future}} \in \mathbb{R}^{H \times W}$, initialized with $-\infty$ and $\infty$, respectively;
7    **for each** $e_n \in E_{prev} \cup E_{future}$ **do**
8       Compute normalized time: $t_{\text{norm}} \leftarrow (t_n - \tau_i)/\tau_{\text{range}}$;
9       **if** $e_n \in E_{prev}$ **then**
10          Update $T_{\text{prev}}(y_n, x_n) \leftarrow t_{\text{norm}}$;
11       **else**
12          Update $T_{\text{future}}(y_n, x_n) \leftarrow t_{\text{norm}}$;
13    **for each** $(x, y) \in \{(x, y) \mid x \in [0, W-1], y \in [0, H-1]\}$ **do**
14       Update $L[i, y, x]$:

$$L[i, y, x] \leftarrow \begin{cases} T_{\text{prev}}(y, x), & \text{if } T_{\text{prev}}(y, x) \neq -\infty \\ T_{\text{future}}(y, x), & \text{if } T_{\text{prev}}(y, x) = -\infty \text{ and } T_{\text{future}}(y, x) \neq \infty \\ -1, & \text{otherwise} \end{cases}$$

15 **return** $L$;

---

Furthermore, the flexibility of Labits enables its use with varying numbers of probe time points and time bin sizes, without the need to retrain the model used to predict instantaneous optical flows. This adaptability stems from the fact that corresponding APLOF are generated separately for each Labits layer, and variations in time bin duration just introduce a scaling factor to the perceived speed. Subsequent deep learning models can readily adjust to any scaling effects introduced by variations in the generated local optical flows. This combination of adaptability and precise flow prediction enhances Labits' utility as a versatile and powerful tool for event-based computer vision tasks.

In conclusion, Labits overcomes the limitations of existing event representations by providing a highly accurate, temporally precise, and robust way to capture motion information from event streams. Its ability to handle both near-past and near-future events, coupled with its stability against time normalization, allows it to deliver superior performance in dense trajectory estimation. By fully harnessing the hardware advantages of event cameras, Labits sets a new benchmark for event-based continuous dense trajectory estimation.

## 4 METHOD

### 4.1 LABITS-TO-APLOF NET

Event cameras are highly sensitive to motion, particularly instantaneous or local motion. In this context, "local" refers to both spatial and temporal locality, aspects not captured by conventional optical flow techniques. For instance, in the MultiFlow dataset, the optical flow ground truth at each intermediate probe time is relative to the initial reference time point, representing cumulative motion rather than instantaneous speed. In contrast, events triggered by the same moving object within a short time frame are closely tied to the object's instantaneous speed, based on the variation in their

timestamps, rather than cumulative displacement. Therefore, to fully leverage event cameras for tracking, local speed must be taken into account.

Labits retains substantial local motion information at active pixels both before and after the probe time points. Compared to voxel grids, the information encoded in Labits enables the main model to predict trajectories with greater accuracy. However, due to the sparsity of events, noise, and temporal occlusion, it can be challenging for a downstream task model to effectively decode this implicit information without any dedicated guidance.

To fully leverage Labits, we propose a general-purpose model component named the Labits-to-APLOF Net, where **APLOF** stands for "**A**ctive **P**ixel **L**ocal **O**ptical **F**low." This model component is specifically designed to utilize a single Labits layer as input for three main reasons: First, each Labits layer, generated at an intermediate probe time point, already contains sufficient information to infer the corresponding APLOF. Second, as we mainly focus on spatiotemporal locality, using a single layer and a shallow network naturally preserves that while keeping the model efficient and enhancing generality. Finally, since the number of time bins in Labits can be freely adjusted by the user, maintaining a single-channel input increases flexibility. This design makes the Labits-to-APLOF Net a versatile module compatible with Labits.

The Labits-to-APLOF Net is implemented based on U-Net (Ronneberger et al. (2015)). To improve both stability and efficiency, we employ instance normalization within each block, rather than batch normalization. This adaptation arises from shifting the channel dimension of Labits to the batch dimension during inference, enabling the network to more effectively handle Labits while keeping the single input channel. Since Labits contains rich information primarily around active pixels, local speed estimation is most reliable in these areas. Thus, we apply an active pixel mask (APM) to the generated APLOF features, considering only Labits pixels with absolute values below a certain threshold $\beta$ as active pixels. To ensure alignment between the low-resolution (LR) APLOF features at the bottleneck and the pixel coordinates, we incorporate an auxiliary output head during training. This head is trained to predict an LR version of the local speed flow based on the bottleneck features.

Let $\hat{\mathbf{A}}_h$ and $\hat{\mathbf{A}}_l$ represents the estimated high-resolution (HR) and LR APLOF, while $E$ and $D_h$ denote the encoder and decoder of the U-Net, respectively. $D_l$ means the auxiliary LR decoder, and $\mathbf{M}_h, \mathbf{M}_l$ stand for HR/LR active pixel masks. The following equation formalizes the process:

$$\mathbf{M}_h = \mathbf{1}_{|L|<\beta} \tag{1}$$

$$\mathbf{M}_l = \mathbf{1}_{\text{AvgPool}_{8\times 8}(\mathbf{M}_h)<\gamma} \tag{2}$$

$$\hat{\mathbf{A}}_h = D_h(E(\mathbf{L})) \cdot \mathbf{M}_h \tag{3}$$

$$\hat{\mathbf{A}}_l = D_l(E(\mathbf{L})) \cdot \mathbf{M}_l \tag{4}$$

In these equations, the term $\mathbf{1}_{|\mathbf{L}|<\beta}$ is an active pixel mask that is generated based on the condition $|\mathbf{L}| < \beta$, where the mask $M_h$ takes the value 1 when the condition is satisfied and 0 otherwise. In this way, pixels where there are events happened most recently stand out. Similarly, the low-resolution active pixel mask is generated based on a down-sampled version of high-resolution active pixel mask, with a threshold of $\gamma$. After fine-tuning, the thresholds $\beta$ and $\gamma$ are set to 0.3 and 0.125.

During training, the ground truth APLOF at $\tau$ is generated using the ground truth optical flow at $\tau_-, \tau$, and $\tau_+$, where $\tau_-, \tau_+$ represents the timestamp of $\tau \pm 10ms$. Denoting a pixel's coordinate at $\tau_{\text{start}}$ as $\mathbf{x}_{\text{start}}$, and optical flow between $\tau_{\text{start}}$ and $\tau$ as $O_\tau$, the ground truth HR APLOF value is:

$$\mathbf{x}_\tau = \mathbf{x}_{\text{start}} + O_\tau(\mathbf{x}) \tag{5}$$

$$\mathbf{A}_\tau(\mathbf{x}_\tau) = (O_{\tau_+}(\mathbf{x}_{\text{start}}) - O_{\tau_-}(\mathbf{x}_{\text{start}})) \cdot \mathbf{M}_h(\mathbf{x}) \tag{6}$$

The training loss combines the LR and HR APLOF losses as follows:

$$\mathcal{L}_A = \|\hat{\mathbf{A}}_h - \mathbf{A}_h\|_1 + \|\hat{\mathbf{A}}_l - \mathbf{A}_l\|_1 \tag{7}$$

### 4.2 PIPELINE

Accurately estimating pixel trajectories over time goes beyond simply determining their final displacements. Our proposed solution aims to predict each pixel's shifting at various intermediate probe times within the defined time range. The RAFT-based structure (Teed & Deng (2020)) excels

*(a) Labits-RAFT Model Structure*       *(b) Labits-to-APLOF Net*

Figure 2: (a) Labits-RAFT architecture: Labits are used to generate correlation blocks, content features, APLOF features for intermediate movement integration, and guide Active Pixel Mask (APM) generation. APM layers are point-wise multiplied to their corresponding APLOF feature layers. Features are used to calculate correlation matrix, eventually generate and refine a Bézier curve **B** for each pixel at $\tau_{\text{start}}$ via a ConvGRU. For brevity, we only show the pure event-based pipeline. (b) Labits-to-APLOF Net: HR and LR APLOF are generated based on a customized U-Net and APM.

at extracting correlations between pixels across different spatiotemporal locations. Predicting the trajectory as a Bézier curve, rather than as a series of discrete displacements, allows for a smoother and more continuous representation. Building on top of (Gehrig et al. (2024)), our pipeline consists of **four main components**: feature extraction, correlation computation, Bézier parameter refinement, and upsampling. The task is mathematically defined by the following equations:

$$\mathbf{B} : \mathcal{T} \times \mathbb{N}_0 \times \mathbb{N}_0 \to \mathbb{R}^2 \tag{8}$$

$$(\tau, x, y) \mapsto B(\tau, x, y) \tag{9}$$

$\mathcal{T} = \mathbb{R} \cap [0, 1]$ describes the domain of the normalized time $\tau(t) = (t - t_r)/(t_t - t_r)$, with $\tau = 0$ and $\tau = 1$ corresponding to the reference time $t_r$ and target time $t_t$, respectively, indicating the start and end of the pixel trajectory. This formulation allows the trained model to predict pixel displacement for any initial pixel at subsequent time points.

**In part one**, the generated event representation and RGB frames (if available) are encoded into feature maps. The time period corresponding to the events is evenly divided into $B$ segments, each lasting milliseconds. Of these, the first $M - 1$ segments provide additional reference information, while the remaining $B - M + 1$ segments constitute the target time period, referred to as the context Labits block. If available, the RGB frame at the reference time is concatenated to the context Labits block to enrich the feature set. Using a sliding window approach, the total $B$ Labits layers are grouped into $J$ correlation Labits blocks, each containing $M$ layers. Each block is also referred to as a "view," as it represents a distinct timestamp.

**In part two**, correlation volumes are computed between the first view and subsequent views to facilitate the spatial-temporal correspondence search for initial pixels. Additionally, if RGB frames are provided, a correlation volume between the reference and target frames is also calculated. These volumes are then utilized to iteratively refine the Bézier parameters.

**In part three**, the goal is to find a set of Bézier control points, $\mathcal{P}$, for each initial pixel at the reference time, $t_r$. Initially, all control points are set to zero, positioning each pixel's default trajectory as a straight line along the time axis. This part primarily involves a ConvGRU, which processes the context and correlation information, along with the Bézier parameters, and outputs the residuals of the Bézier parameters. The initial hidden states of the ConvGRU are generated from the context volumes and APLOF features. During each subsequent iteration, the Bézier parameters are progressively refined to achieve optimal estimation. The Bézier curve is defined as follows:

$$\mathbf{B}(\tau, x, y) = \sum_{i=0}^{n} \binom{n}{i} (1 - \tau)^{n-i} \tau^i \mathbf{P}_i(x, y) \tag{10}$$

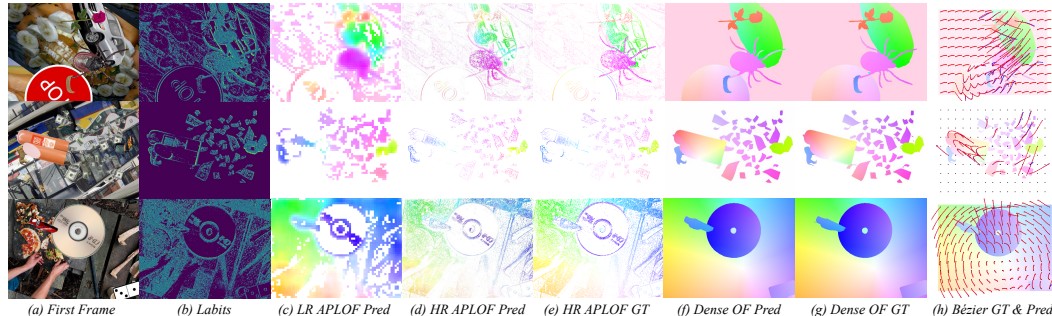

(a) First Frame    (b) Labits    (c) LR APLOF Pred    (d) HR APLOF Pred    (e) HR APLOF GT    (f) Dense OF Pred    (g) Dense OF GT    (h) Bézier GT & Pred

Figure 3: Visualization of detailed inputs and outcomes from our model. It predicts instantaneous APLOF at intermediate reference times, end-point optical flow (the end-point optical flow is computed as the displacement of each pixel along its entire trajectory), and pixel-level Bézier trajectories, all closely aligning with the corresponding ground truth data. *OF: Optical Flow.

Here, $\mathcal{P}$ represents the set of Bézier control point parameters, with $\mathcal{P} = \mathbf{P}_1, \ldots, \mathbf{P}_n$ when the Bézier curve has $n$ degrees. After the iterative refinement of the low-resolution Bézier parameters, convex upsampling is applied to upscale the Bézier parameters to the original resolution (**part four**).

The model is supervised with $N_k$ ground truth optical flows along the trajectory. Let $\mathbf{B}_i$ denote the Bézier curve at iteration $i$, $\tau_k \in [0, 1]$ represent the evaluation timestamps, and $\gamma = 0.8$. The loss function is defined as follows, with more details provided in (Gehrig et al. (2024)).

$$\mathcal{L} = \frac{1}{N_k} \sum_{i=1}^{N_i} \gamma^{N_i - i} \sum_{k=1}^{N_k} \|\mathbf{f}_{gt}(T_k) - \mathbf{B}_i(T_k)\|_1 \tag{11}$$

APLOF features introduced earlier are utilized in parts one and three. In part one, APLOF features are generated using the selected Labits layers that correspond to the intermediate trajectory ground truth time points. These features are then merged with the associated correlation block features based on the time range of each view. In part three, the initial hidden states of the ConvGRU, generated from the context block, are enhanced by merging with APLOF features. This integration provides additional local motion information, significantly improving trajectory estimation accuracy. The results section demonstrates the exceptional effectiveness of the APLOF features.

## 5 RESULTS

**Datasets and Evaluation Metrics.** Our model is trained and evaluated on the MultiFlow dataset (Gehrig et al. (2024)), which comprises 10,100 training and 2,000 test sequences, each including paired RGB images, events, and optical flows relative to a reference time. With optical flow ground truths recorded every 10 ms, we can generate intermediate local optical flows for training the Labits-to-APLOF net, and ensure accurate supervision of the intermediate Bézier parameters. MultiFlow is currently the only available event dataset that supports dense trajectory prediction and active pixel local optical flow estimation simultaneously. The ground truth for pixel trajectories is provided relative to a reference time of 0.4 seconds ($t_{ref}$), with trajectories calculated at 10-millisecond intervals between $t_{ref}$ (0.4 seconds) and $t_{tar}$ (0.9 seconds). Events occurring outside this range provide additional context. To evaluate the trajectory-level estimation accuracy, beyond the standard EPE and AE metrics, we also adopt Trajectory End-Point Error (TEPE) and Trajectory Angular Error (TAE).

**Labits-to-APLOF Net Results.** The Labits-to-APLOF Net is supervised by high- and low-resolution APLOF ground truth calculated via the Equation 6, with a single Labits layer as its basic input unit. This lightweight network (0.524M parameters) efficiently captures local movement information from Labits layers, exhibiting high-quality APLOF estimation. In the main trajectory estimation pipeline, only the encoder part is used, reducing the parameter count to 0.293M. The Labits-to-APLOF net is trained using a StepLR scheduler for 60 epochs and 75k iterations. Training on an NVIDIA RTX 3090 GPU took 35 hours, achieving a low error rate. The model converged with a minimal total loss (Equation 7) of 0.084, consisting of LR APLOF L1 loss (0.056) and HR APLOF L1 loss (0.028). Additional qualitative results are detailed in the Supplementary Material.

**Main Pipeline Results.** Figure 3 presents the visualization of our method's results on the Multi-Flow dataset. As shown, both the HR and LR versions of the predicted APLOF closely align with the corresponding ground truth in the early stages, benefiting the main model and offering additional guidance. For applications that require only the optical flow of active pixels, early exiting here is feasible. Furthermore, the final pixel trajectories and the resulting dense end-point optical flow accurately align with the ground truth, effectively handling challenging scenarios involving small objects and sharp details. More qualitative results are shown in the Supplementary Material.

Table 2: Results on MultiFlow dataset. TEPE and TAE represent trajectory-based EPE and AE. Metrics in parentheses are from a linear motion model, indicating these methods aren't optimized for trajectory prediction. Methods using both events and images are highlighted in grey. Percentage decreases are relative to DCT-RAFT, the top baseline. "w/o" means without.

| Method | Input | Trajectory Metrics | | 2-View Metrics | |
|---|---|---|---|---|---|
| | | TEPE | TAE | EPE | AE |
| RAFT | I | (6.89) | (19.31) | 7.42 | 6.71 |
| RAFT + GMA | I | (5.14) | (16.35) | 1.47 | 1.56 |
| E-RAFT | E | (6.70) | (18.44) | 7.56 | 6.19 |
| E-RAFT+Bézier | E | 2.62 | 5.92 | 4.54 | 6.06 |
| DCT-RAFT | E | 1.85 | 4.61 | 3.37 | 4.80 |
| DCT-RAFT | E+I | 1.29 | 3.35 | 2.27 | 3.19 |
| **Labits-RAFT (Ours)** | E | **1.32 ↓ 29%** | **3.14 ↓ 32%** | **2.50 ↓ 26%** | **3.37 ↓ 30%** |
| **Labits-RAFT (Ours)** | E+I | **0.66 ↓ 49%** | **1.72 ↓ 49%** | **1.08 ↓ 52%** | **1.45 ↓ 55%** |
| Ours w/o APLOF Features | E | 1.53 ↓ 17% | 3.69 ↓ 20% | 2.83 ↓ 16% | 3.89 ↓ 19% |
| Ours w/o APLOF Features | E+I | 1.01 ↓ 22% | 2.63 ↓ 21% | 1.64 ↓ 28% | 2.25 ↓ 29% |
| Ours w/o APLOF & Labits | E | 1.83 | 4.57 | 3.29 | 4.74 |
| Ours w/o APLOF & Labits | E+I | 1.16 | 3.01 | 1.99 | 2.78 |

To evaluate the effectiveness of our approach, we compare it with four previously published baselines: RAFT (Teed & Deng (2020)), RAFT-GMA (Jiang et al. (2021)), E-RAFT (Gehrig et al. (2021b)), and DCT-RAFT (Gehrig et al. (2024)). Specifically, RAFT and RAFT-GMA are frame-based approaches utilizing the RAFT architecture, whereas E-RAFT and DCT-RAFT are recent event-based approaches. DCT-RAFT also has a version that takes both frames and events as input. We select these baselines for comparison because they are all derived from the RAFT framework, providing a consistent basis for evaluating our method's performance.

As presented in Table 2, our method, Labits-RAFT, outperforms other methods across all metrics by a remarkably large margin. Our proposed method reduces the TEPE by 49%, from 1.29 to 0.66, compared to the previous state-of-the-art. Similarly, the Trajectory Angular Error (TAE) is reduced by 49%, from 3.35 to 1.72. These results indicate that approaches directly predicting pixel displacements are not suitable for accurate pixel trajectory estimation (e.g., RAFT, RAFT + GMA, E-RAFT). While incorporating Bézier estimation into E-RAFT improves trajectory and two-view metrics, the estimated flow still has room for improvement. DCT-RAFT, which integrates image frames and introduces correlation features, effectively reduces errors in both trajectory and two-view metrics. However, DCT-RAFT's reliance on voxel grids leads to a significant loss of fine-grained temporal information, representing a critical limitation. Finally, our method also achieves improvements on two-view metrics (52% on EPE and 55% on AE), additionally demonstrating its superiority in accumulative optical flow estimation.

**Ablation Studies.** We conduct ablation studies to assess the contributions of our main innovations in dense continuous-time trajectory estimation: the novel Labits event representation and the Labits-to-APLOF net. We carry out two sets of experiments: one excluding the Labits-to-APLOF network and another further replacing Labits with voxel grids. The results, shown in the last four rows of Table 2, demonstrate significant performance declines without these modules in both pure event and event+image scenarios, confirming the effectiveness of both Labits and APLOF features.

Furthermore, Figure 4 provides qualitative comparisons, showcasing two frames and the pixel-wise movement trajectory between them. Our method is compared with the top baseline methods and key ablated results. The superior curve proximity between our predicted trajectory and the ground truth highlights the benefits of incorporating Labits' implicit local speed cues and APLOF features.

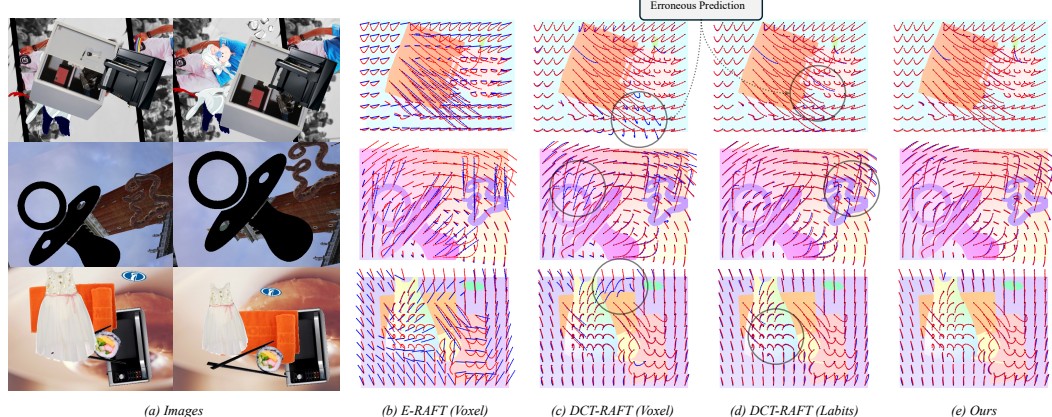

(a) Images  (b) E-RAFT (Voxel)  (c) DCT-RAFT (Voxel)  (d) DCT-RAFT (Labits)  (e) Ours

Figure 4: Trajectory predictions on the MultiFlow dataset by our proposed model and baseline methods. Ground truth Bézier trajectories are shown in red, while predictions are depicted in blue. The background displays the ground truth optical flow to highlight moving objects. Our model's predicted trajectories significantly outperform those of all baseline methods.

**Implementation Details.** We implemented our models in PyTorch, training them on the MultiFlow dataset from scratch with AdamW optimizer (Loshchilov & Hutter (2019)), gradient clipping in the range of [-1, 1], and a OneCycle learning rate scheduler. We adopted trajectory loss and two-view loss as detailed in (Gehrig et al. (2024)), supervising the training with 10 flow maps and employing a Bézier curve of degree 10 for precise trajectory modeling. Our model configuration utilizes $J = 6$ correlation Labits blocks across time intervals from 0.4 to 0.9 seconds, quantizing time into $N = 41$ and $M = 25$ bins for context and correlation, respectively. To ensure consistency in evaluation, we adopt the same setup as described in (Gehrig et al. (2024)). Training spanned 100 epochs and 250k iterations, requiring approximately 40 hours on four NVIDIA GeForce RTX 3090 GPUs.

**Efficiency Analysis.** In our systematic performance evaluation, Labits emerged as an exceptionally efficient representation, adept at capturing dense and detailed motion information with minimal computational delay. We assessed the processing times for various representations, as shown in Figure 1, utilizing randomly sampled 500 event packets from the MultiFlow validation set, with each packet focusing on the initial 100 milliseconds of data. Labits achieved an average execution time of 0.220s, demonstrating comparable efficiency to the Voxel Grid (0.225s) and surpassing the more computationally intensive TORE Volume (7.624s) and Time Surface (0.358s). Notably, simplest Event Frame recorded the fastest execution time at 0.062s, followed by Event Count at 0.102s. All the testing is conducted on the same device under the same condition. Although Labits is not the quickest, it offers a superior balance of speed and data richness compared to the others.

## 6 CONCLUSION

In this work, we introduced Labits, a novel event representation that simultaneously retains fine-grained temporal information, meaningful 2D visual patterns, and local speed cues. Labits is the first to achieve this combination. We also developed the Labits-to-APLOF net, which accurately converts Labits into active pixel local optical flows. Together, these innovations significantly improve performance on event-based continuous-time dense trajectory estimation, with a remarkable 49% reduction in trajectory end-point error compared to the top baseline models. The results highlight that, beyond model architecture, event representations also have a transformative impact on the final outcomes. While Labits' strengths are most apparent in tasks that utilize intermediate motion information, its adaptability to other event-based vision tasks remains an open area for future research. This versatility positions Labits as a valuable asset for advancing event-based vision, offering numerous possibilities for further exploration and development. Detailed analysis of limitations and future research directions could be found in the Supplementary Material.

## REPRODUCIBILITY STATEMENT

The code for our proposed Labits representation and the main model pipeline is packaged in "Labits-Core-Code.zip," available in the supplementary material. This archive encompasses the primary contributions detailed in the paper. Alongside the publication of our paper, we will release a comprehensive GitHub repository containing detailed documentations.

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

# A   SUPPLEMENTARY MATERIAL

*(a) E-RAFT (Voxel)*          *(b) DCT-RAFT (Voxel)*          *(c) Ours w/o APLOF*          *(d) Ours*

Figure 5: Comparison of trajectory predictions: between baseline methods and our approach.

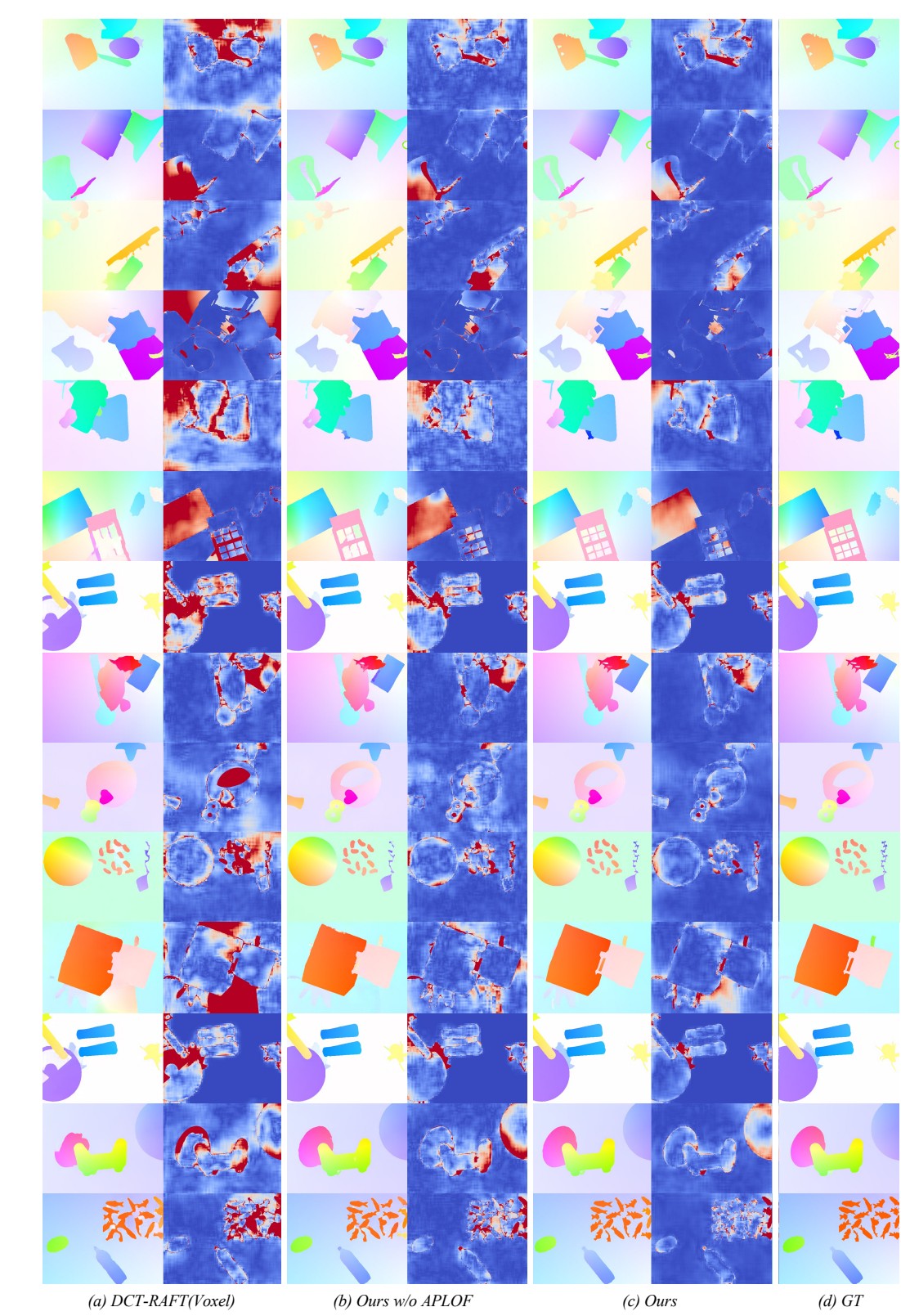

(a) DCT-RAFT(Voxel)    (b) Ours w/o APLOF    (c) Ours    (d) GT

Figure 6: Comparison of Optical Flow Estimations: Each method includes a second column displaying the error map generated by comparing the predictions to the ground truth.

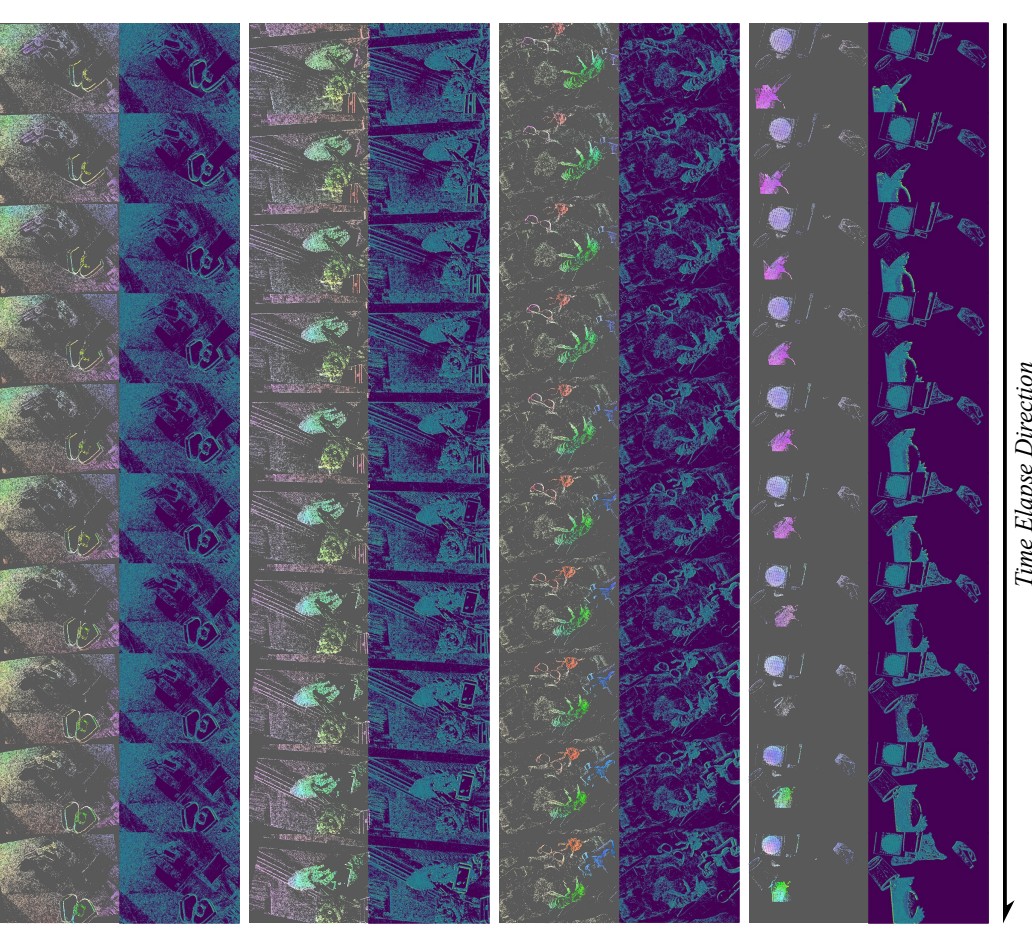

Figure 7: Visualization of APLOF and Labits in Trajectories: For each trajectory, Labits and APLOF are aligned in corresponding pairs across two columns. We selected ten slices of both Labits and APLOF for different trajectories. We employ the "viridis" color map to represent Labits and enhance the visibility of APLOF against a gray background.

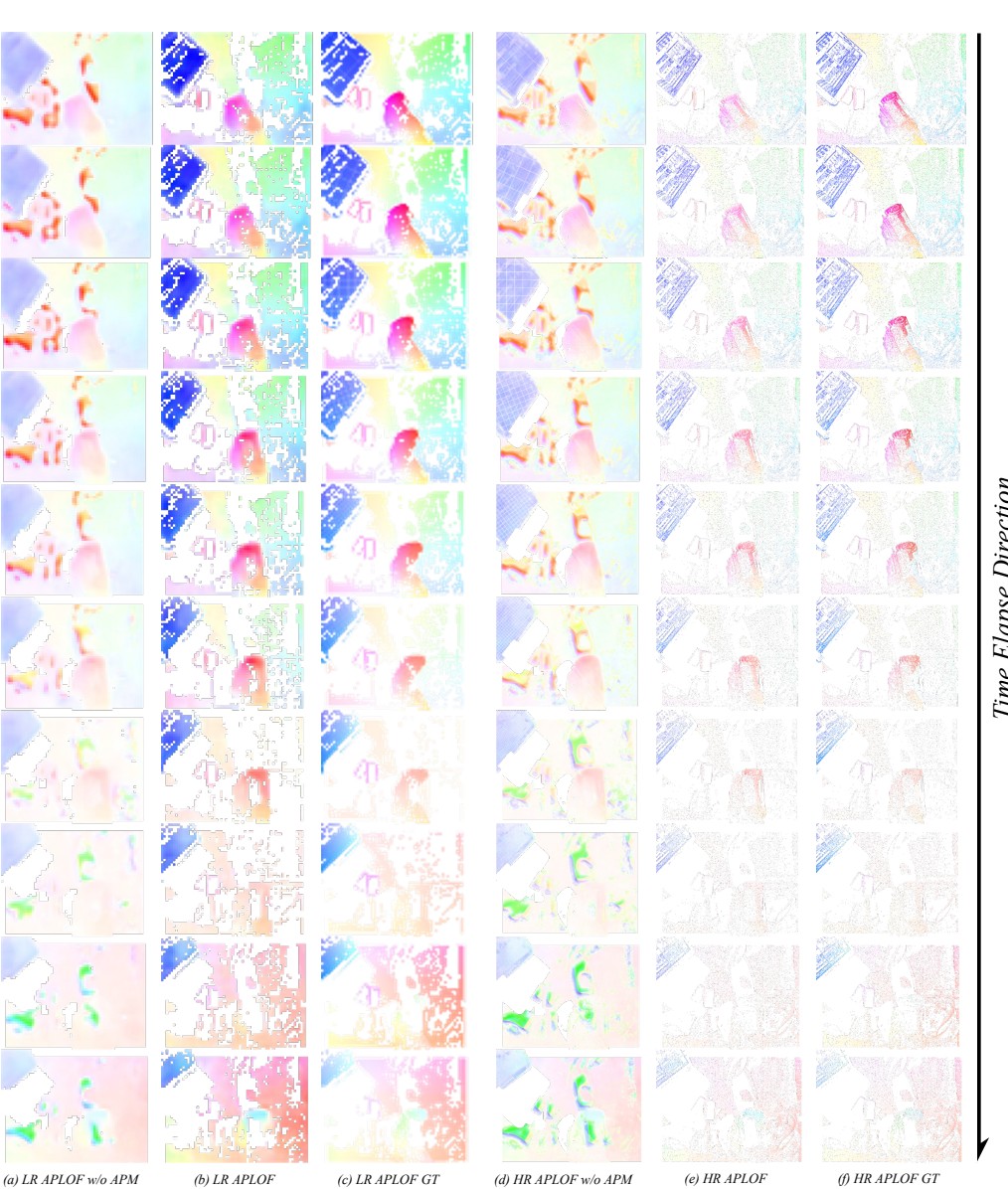

*(a) LR APLOF w/o APM*  *(b) LR APLOF*  *(c) LR APLOF GT*  *(d) HR APLOF w/o APM*  *(e) HR APLOF*  *(f) HR APLOF GT*

Figure 8: Visualization of HR and LR APLOF with APM Ablation: For each trajectory, we display the results of APLOF, APLOF without APM, and the GT across three columns, applicable to both HR and LR contexts. We have visualized ten representative slices from both HR and LR APLOF, juxtaposed with their counterparts lacking APM, to demonstrate the impact of APM ablation on APLOF estimation.

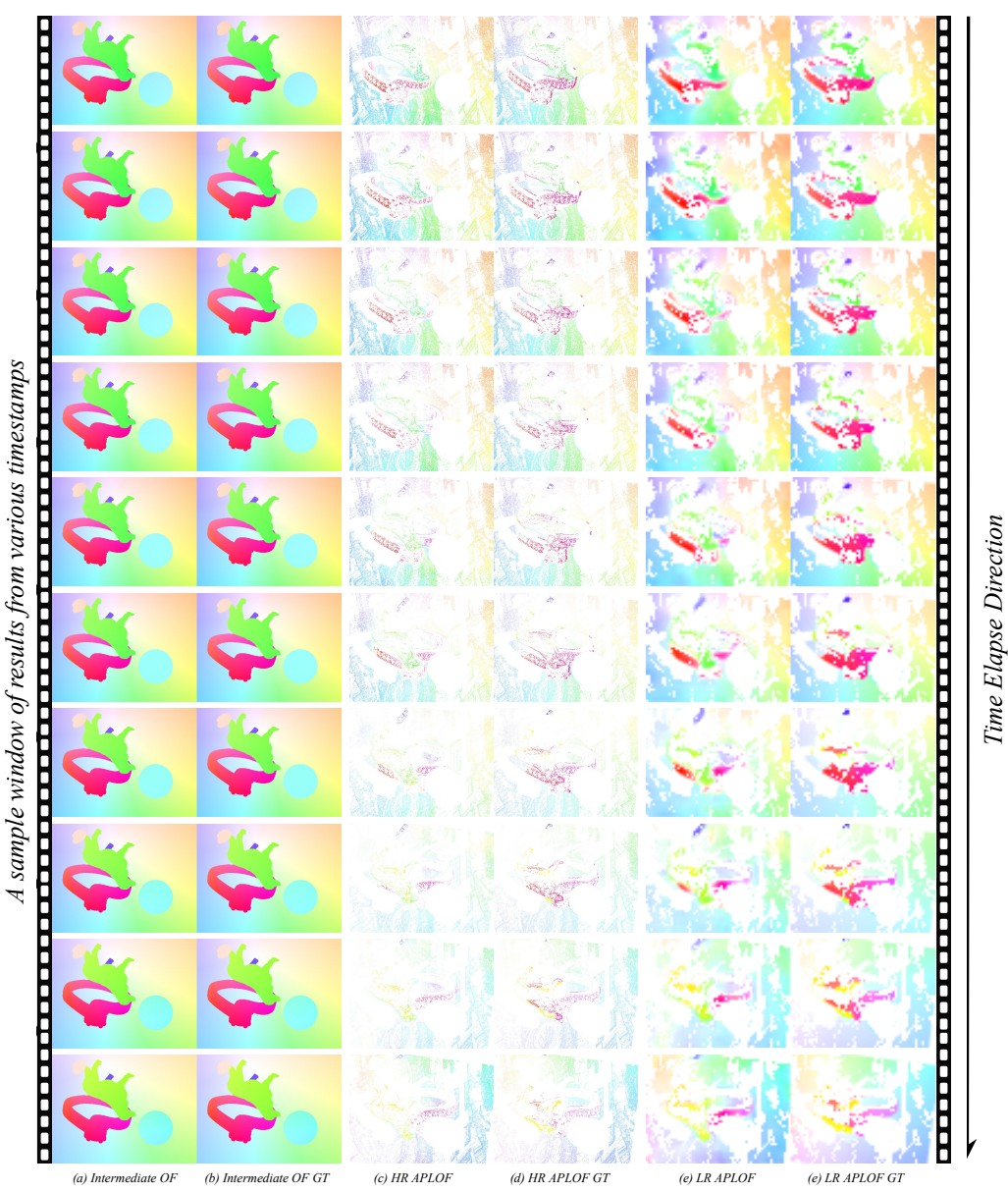

*(a) Intermediate OF*    *(b) Intermediate OF GT*    *(c) HR APLOF*    *(d) HR APLOF GT*    *(e) LR APLOF*    *(e) LR APLOF GT*

Figure 9: Visualization of Intermediate Prediction Results: We display dense, time-continuous prediction outcomes for each trajectory within a sample window from the MultiFlow dataset Gehrig et al. (2024). The visualization includes ten intermediate instances of optical flow (OF), HR APLOF, LR APLOF, and the corresponding GT, presented in three paired columns. Timestamps for the predictions correspond to those of the GT. The trajectory spans a time range of 0.5 seconds (context). We evenly sampled 10 intermediate timestamps, and show the predicted and groud truth OF between them and the start time. Meanwhile, APLOFs show instantaneous optical flows.

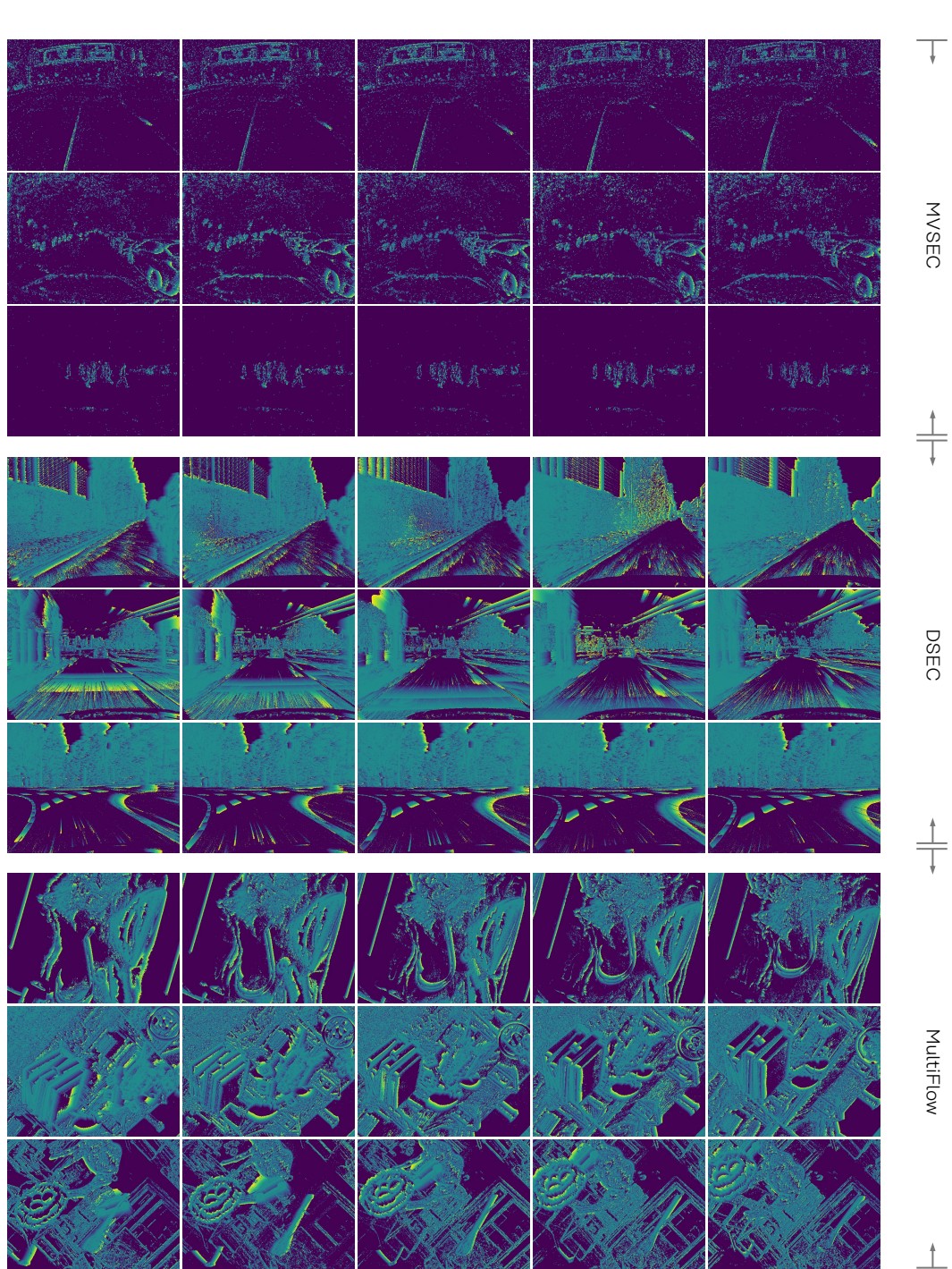

Figure 10: Visual Comparison of Labits in Various Scenarios: We generate Labits on three well-known event camera datasets: DSEC Gehrig et al. (2021a), MVSEC Zhu et al. (2018a), and Multi-Flow Gehrig et al. (2024). The time ranges and the number of bins are adjusted to accommodate the data formats of each dataset and to achieve better visual clarity.

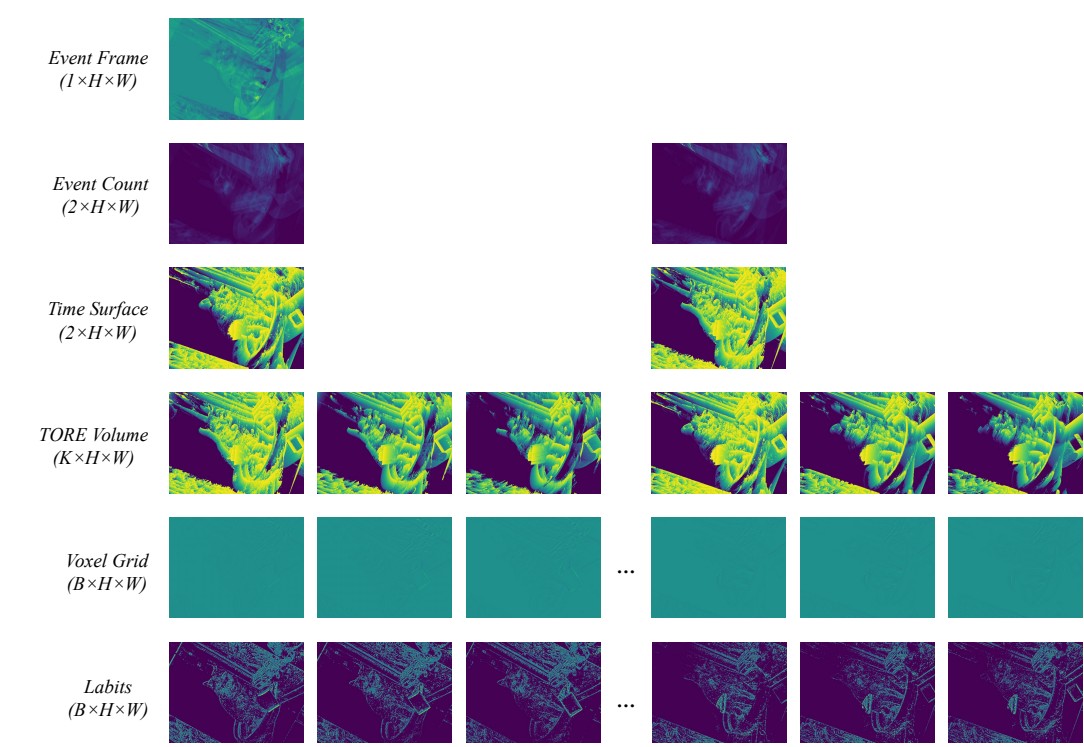

Figure 11: Comparative Visualization of Event Representations: This figure presents event data sampled between 100 ms and 900 ms from the MultiFlow dataset, showcasing diverse representations such as Time Surface, Event Frame, TORE Volume (K=3), Voxel Grid (B=65), and Labits (B=65). These hyperparameters are selected following the corresponding paper's settings. Each method distinctively captures and structures event information. The first three plots of TORE Volume correspond to positive events' representation layers, while negative for the last three. Voxel Grid and Labits visualized the first three and last three bins, respectively. For time surface and event count, the first and second frames represent the count of negative and positive events.

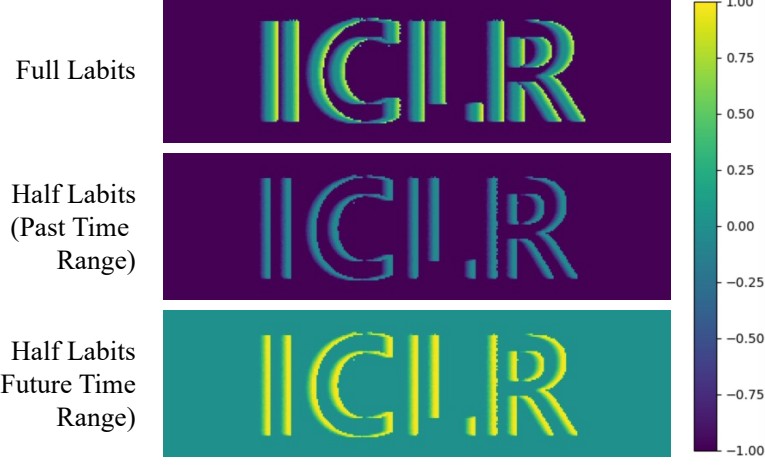

Figure 12: Visual Comparison of Labits and One-way Labits: We visually compare Labits and One-way Labits generated from the same icon sample, as presented in Figure 1. For the One-way Labits, we create results using only the past or future time ranges, respectively, to illustrate the visual difference.

## A.1  Model Specifications

Table 3: DCT-RAFT and Labits-RAFT Model Size and Performance Metrics

| Metric | DCT-RAFT | Labits-RAFT* | Labits-RAFT |
|---|---|---|---|
| APLOF Features | N/A | Yes | Yes |
| Trainable Parameters | 6.8 M | 6.8 M | 24.8 M |
| Non-trainable Parameters | 0 | 0 | 524 K |
| Total Parameters | 6.8 M | 6.8 M | 25.3 M |
| Estimated Model Size | 27.203 MB | 27.203 MB | 101.288 MB |
| Average GFLOPS | 4316.287 | 4432.353 | 5599.052 |
| Average Inference Time | 0.133 s | 0.135 s | 0.192 s |
| TEPE | 1.29 | 1.01 | 0.66 |
| TAE | 3.35 | 2.63 | 1.72 |

Table 3 provides detailed performance metrics for the DCT-RAFT and Labits-RAFT models (* means Labits-RAFT w/o APLOF features), based on 100 iterations on an NVIDIA-A10 GPU. The DCT-RAFT model, with 6.8 million parameters, delivers fast inference times but higher error rates, while the more complex Labits-RAFT, with 25.3 million parameters, exhibits a slight increase in inference time but significantly reduces errors by 49%, illustrating the trade-offs between model complexity and performance efficiency.

## A.2  Instantaneous Speed Prediction Error Estimation Analysis

The motivation behind the development of Labits stems from the necessity of accurately estimating continuous dense optical flow—a process that requires precise understanding of instantaneous pixel-level movements at numerous intermediate moments within a scene. These movements information, which we refer to as local motion anchors, play a crucial role in tasks that demand detailed motion understanding. Based on the formula of the backward (Eq.12) and central difference (Eq.13), combined with Taylor expansion (Eq.14, Eq.15), we know the error estimation for backward difference is $O(\delta t)$, while the one for central difference is $O(\delta t^2)$, meaning central difference provides higher accuracy.

$$f'(x_n) \approx (f(x_n) - f(x_{n-\delta t}))/\delta t \tag{12}$$

$$f'(x_n) \approx (f(x_{n+\delta t}) - f(x_{n-\delta t}))/2\delta t \tag{13}$$

$$f(x_{n+\delta t}) = f(x_n) + \delta t f'(x_n) + 0.5(\delta t)^2 f''(x_n) + O((\delta t)^3) \tag{14}$$

$$f(x_{n-\delta t}) = f(x_n) - \delta t f'(x_n) + 0.5(\delta t)^2 f''(x_n) - O((\delta t)^3) \tag{15}$$

In each Labits layer, the values represent relative time information, enabling the scaled instantaneous local speed to be calculated as the inverse of the 2D Labits gradient on the image sensor plane. To improve local speed estimation at active pixels, the central difference method, which requires relative time values from both the past and future, is preferred. Therefore, the Labits implementation incorporates both near-past and near-future events at each reference point. In contrast, existing time surface-style representations, which compress information solely from the past, are less accurate for local speed estimation and leave a significant portion of pixels in front of moving objects empty, reducing information density, especially when the time range is short.

## A.3  Analysis of the Bidirectional Structure in Labits

Since Bidirectional is a key feature of our event representation, this section can provide more concrete evidence to show the effectiveness of this design scheme. We conduct additional ablation studies on whether the representation takes in bidirectional information during the representation generation. For the so-called "One-way Labits", only events triggered during the past time range is considered, following the traditional time surface generation strategy. We retrain the entire pipeline using the "One-way Labits", including the Labits-to-APLOF net and the Labits-RAFT model. The results of both model is significantly worse compared to our proposed bidirectional Labits. Details

are shown in Table A.2 and Table 5. Additionally, visual comparison of Labits and One-way Labits are detailed in Figure 12.

Table 4: Comparison of APLOF Losses

| Model | Total Loss | LR APLOF L1 Loss | HR APLOF L1 Loss |
|---|---|---|---|
| Labits-to-APLOF Net | 0.084 | 0.056 | 0.028 |
| Labits-to-APLOF Net (One-way) | 0.357 | 0.199 | 0.158 |

Table 5: Performance Metrics of Model Pipelines

| Method | Input | TEPE | TAE | EPE | AE |
|---|---|---|---|---|---|
| DCT-RAFT | E+I | 1.29 | 3.35 | 2.27 | 3.19 |
| Labits-RAFT (Ours) | E+I | 0.66 | 1.72 | 1.08 | 1.45 |
| Labits-RAFT (One-way) | E+I | 0.72 | 1.87 | 1.22 | 1.68 |

### A.4 ABLATION STUDY ON TIME BIN SIZE

Besides, we conduct ablation studies on the number of Probe Times to facilitate comprehensive evaluations. Detailed ablation studies focus on different bin configurations for Labits and Voxel Grid, specifically in the context of trajectory estimation. These studies utilize the same pretrained Labits-to-APLOF network, which was originally trained with Labits featuring 65 bins and a bin size of 0.0125 seconds. We design our study to cover a variety of scenarios by setting bin time spans at 0.0125s, 0.0250s, 0.0500s, and 0.1000s, facilitating a direct comparison with DCT-RAFT Gehrig et al. (2024). Table 6 provides robust results that thoroughly validate the efficacy of Labits, confirming that the pretrained Labits-to-APLOF net can adaptively handle different time bins.

Table 6: Performance comparison of DCF-RAFT and Labits-RAFT using different bin setups.

| Model | Bin Size | TEPE | TAE | EPE | AE |
|---|---|---|---|---|---|
| DCF-RAFT (voxel) | 0.1000 | 1.81 | 4.53 | 2.87 | 3.97 |
| **Labits-RAFT (Labits)** | **0.1000** | **0.88 ↓ 51%** | **2.35 ↓ 48%** | **1.43 ↓ 50%** | **1.97 ↓ 50%** |
| DCF-RAFT (voxel) | 0.0500 | 1.51 | 3.82 | 2.61 | 3.63 |
| **Labits-RAFT (Labits)** | **0.0500** | **0.75 ↓ 50%** | **1.94 ↓ 49%** | **1.22 ↓ 53%** | **1.66 ↓ 54%** |
| DCF-RAFT (voxel) | 0.0250 | 1.40 | 3.56 | 2.45 | 3.41 |
| **Labits-RAFT (Labits)** | **0.0250** | **0.72 ↓ 49%** | **1.86 ↓ 48%** | **1.17 ↓ 52%** | **1.58 ↓ 54%** |
| DCF-RAFT (voxel) | 0.0125 | 1.29 | 3.35 | 2.27 | 3.19 |
| **Labits-RAFT (Labits)** | **0.0125** | **0.66 ↓ 49%** | **1.72 ↓ 49%** | **1.08 ↓ 52%** | **1.45 ↓ 55%** |

### A.5 ADDITIONAL COMPARISON EXPERIMENTS

To further validate the performance of the Labits-RAFT model, we train a Voxel-to-APLOF net under the same conditions as the Labits-to-APLOF net. For active pixels, we emply a zero mask. Additionally, we implemented a complete model pipeline using voxel and the APLOF feature generated by the pretrained Voxel-to-APLOF net. Table 7 provides a comparative overview of the minimal total loss (Equation 7) between the Labits-to-APLOF net and the Voxel-to-APLOF net. Furthermore, Table 8 details the performance metrics of the complete model pipeline that incorporates the Voxel-to-APLOF features and voxel.

Furthermore, we deploy event representations that are computationally feasible and align well with the layered representation requirements. In this context, we implement and test the Former-Latter

Table 7: Comparison of APLOF Losses

| Model | Total Loss | LR APLOF L1 Loss | HR APLOF L1 Loss |
|-------|-----------|------------------|------------------|
| Labits-to-APLOF Net | 0.084 | 0.056 | 0.028 |
| Voxel-to-APLOF Net | 0.217 | 0.080 | 0.137 |

Table 8: Performance Metrics of Model Pipelines

| Method | Input | TEPE | TAE | EPE | AE |
|--------|-------|------|-----|-----|-----|
| DCT-RAFT | E+I | 1.29 | 3.35 | 2.27 | 3.19 |
| Labits-RAFT (Ours) | E+I | 0.66 | 1.72 | 1.08 | 1.45 |
| DCT-RAFT + Voxel-to-APLOF features | E+I | 0.84 | 2.25 | 1.40 | 1.97 |

Event Groups (Lee et al. (2020)) and Gaussian Weighted Polarities (Ding et al. (2022)), as shown in 9. These methods fit within our computational constraints and ensure fair comparisons across different tasks. This comparison not only facilitates a thorough evaluation of each representation's efficacy under consistent conditions but also validates the effectiveness of Labits.

Table 9: Comparison of more recent layered event representations applied to the RAFT architecture (Teed & Deng (2020)) for trajectory estimation.

| Method | Input | TEPE | TAE | EPE | AE |
|--------|-------|------|-----|-----|-----|
| Former-Latter Event Groups | E+I | 3.15 | 7.36 | 3.81 | 5.07 |
| Gaussian Weighted Polarities | E+I | 3.47 | 8.16 | 4.23 | 5.46 |
| DCT-RAFT | E+I | 1.29 | 3.35 | 2.27 | 3.19 |
| Labits-RAFT (w/o APLOF) | E+I | 1.01 | 2.63 | 1.64 | 2.25 |
| **Labits-RAFT (Ours)** | **E+I** | **0.66 ↓ 34.7%** | **1.72 ↓ 34.6%** | **1.08 ↓ 34.2%** | **1.45 ↓ 35.6%** |

## A.6    LIMITATIONS AND FUTURE DIRECTIONS

### A.6.1    LIMITATIONS

**The proposed solution has specific use cases.** Labits demonstrates excellent performance in dense continuous-time trajectory estimation tasks. However, this is not only due to our novel event representation and a well-designed model pipeline, but also because the task definition and the selected dataset inherently leverage Labits' ability to preserve rich intermediate temporal information. In this task, where continuous motion trajectories of each pixel over the target time span are evaluated, and these intermediate states are included in the metrics, our solution significantly improves results. Conversely, for tasks less sensitive to intermediate states and focused only on final outcomes, such as object detection or optical flow estimation, Labits may not outperform some well-established representations and solutions. In these cases, the additional fine-grained intermediate information may even become irrelevant noise. Different tasks require different types of information, and while our solution is not a one-size-fits-all approach, it can exhibit substantial advantages for suitable tasks.

**Labits omits event density information.** Labits is designed to maximize the extraction and retention of motion information at intermediate time points. However, this design sacrifices event density information, which might be crucial for certain computer vision tasks. Future work could explore combining multiple representations to create a more generalizable framework that balances these trade-offs.

**We did not conduct extensive adaptation experiments across various event camera-based vision tasks.** This paper focuses on providing an exceptional solution for the specific task of dense continuous-time trajectory estimation, rather than proposing a universal representation. Applying

Labits is not as simple as directly replacing previously used representations in various tasks. Instead, it requires tailored feature fusion designs for Labits, APLOF, and the tasks themselves. Redesigning and conducting training, testing, and ablation studies for diverse tasks is beyond the scope of a single conference paper and was not our intention. This work remains centered on the specific task highlighted in the title.

### A.6.2 FUTURE RESEARCH DIRECTIONS

This paper introduces Labits along with a series of models and structures that fully exploit its capabilities. We have demonstrated its unparalleled suitability for dense continuous-time trajectory estimation and validated the effectiveness of Labits and its associated structures through comprehensive ablation studies. Potential future research directions include:

**Expanding Labits Applications to Diverse Tasks** Adapting Labits and its associated models to other similar tasks, such as human keypoint tracking, object trajectory tracking, or event-based video interpolation. These adaptations would undoubtedly require redesigned model structures tailored to each task's specific needs.

**Targeted Tracking Focused on Active Pixels** Leveraging the properties of active pixels proposed in this work to generate representations for selective regions rather than globally. This approach could enable more efficient tracking by focusing on active pixel clusters.

**Refinement of Labits** Combining Labits with traditional event representations like Voxel Grid to achieve complementary strengths. Labits captures fine-grained temporal information for extracting local motion states at intermediate moments but lacks event density information. Conversely, Voxel Grid provides event density information but discards the fine-grained temporal precision that event cameras excel at. Combining these representations may open new possibilities for many tasks.

**Utlizing Polarity Information** Labits, for efficiency and information density considerations, does not differentiate between events of different polarities. Exploring more optimized handling of polarity-specific events may further enhance its performance in relevant computer vision tasks.

In conclusion, the innovations proposed in this paper are versatile and can be extended or improved upon. We hope more researchers will contribute to developing better event representations, unlocking the full potential of event cameras.

