# OpenReview forum: "Labits: Layered Bidirectional Time Surfaces Representation for Event Camera-based Continuous Dense Trajectory Estimation"
_ICLR.cc/2025/Conference — Submitted to ICLR 2025_

### Official Review · Reviewer_nUyE · 2024-10-18

**Soundness:** 3
**Presentation:** 3
**Contribution:** 3
**Rating:** 6
**Confidence:** 4

**Summary:**

This paper introduces a simple and elegant representation to preserve fine-grained event features for event-based continuous dense trajectory estimation. Experiments conducted on the MultiFlow dataset demonstrate the effectiveness of the proposed solution. Extensive analyses and visualizations are presented. The source code will be provided to foster future research.

**Strengths:**

1. A novel and effective synchronous event representation aware of the event camera's asynchronous characteristics is presented.
2. The paper is overall well-written and nicely structured.
3. The source code will be provided to foster research in this direction.
4. Rich interesting visualization results demonstrate the effectiveness of the proposed solution.

**Weaknesses:**

1. If it is possible, it would be nice to investigate the potential of the proposed representation for optical flow estimation or mesh flow estimation.
2. In addition to the MultiFlow dataset, it would be nice to experiment on another dataset to show the generalization capacity of the proposed solution.
3. If it is possible, please consider presenting analyses to directly show the dense time-continuous prediction results by visualizing the results in different time steps within a sample window of the MultiFlow dataset.
4. More detailed ablation studies and parameter analyses could be conducted. For example,  the numbers of the TORE volume, the voxel grid, the Labits, and time bins could be experimented for analyzing the best parameter settings or helping understand the effects of different parameters.

**Questions:**

1. Would you consider discussing some limitations of the presented work and pointing out some future research directions?
2. Would you consider directly comparing the proposed representation against some existing event representations to verify the superiority of your solution?
3. In Table 1, it is indicated that the voxel grid discards the event polarity. However, voxel grids actually contain the accumulation of event polarity information. Could you explain this?
4. Actually, for voxels, they can also support the training of a voxel-to-APLOF network by computing a voxel active event mask and computing voxels with accumulative time spans, similar to the unified voxel grid of EVA-Flow. Could you explain this?
5. What is r_start in equation(5)(6)? Does O(x) indicate the optical flow speed or displacement? Then how did you compute Or(x), Or+(x_start), and Or-(x_start)? This is somewhat misleading.
6. It is claimed that "Labits is well-suited for local motion-sensitive tasks and has the potential to standard". However, only the last two rows in Table 2 indicate the results of Labits. It would be nice to add more results to verify the superiority of Labits, e.g., by testing DCT-RAFT with Labits and by testing with Unified Voxel Grid.
7. While the proposed method achieves high accuracy on MultiFlow, it is a synthetic dataset. It would be nice to offer results on a real-world dataset like DSEC-Flow to verify the generalization capacity of the proposed method.

---

> ### Author Response · Authors · 2024-11-22
> **Q1: Discussion on Limitations and Future Directions**
>
> Dear Reviewer nUyE,
>
> Thank you for your time and valuable suggestions. We appreciate your recognition of our research's practical value and contributions. We aim to address your comments and provide additional information to clarify our work's strengths and implications. Due to the large amount of content, the reply will be divided into several parts.
>
> ## Q1: Discussion on Limitations and Future Directions:
>
> **reponse** : Thank you for emphasizing the importance of discussing the limitations and future directions of our study. We have identified the limitations of our work and outlined potential future directions.
>
> ### Limitations
>
> 1. **The proposed solution has specific use cases.**
>    Labits demonstrates excellent performance in dense continuous-time trajectory estimation tasks. However, this is not only due to our novel event representation and a well-designed model pipeline but also because the task definition and the selected dataset inherently leverage Labits’ ability to preserve rich intermediate temporal information. In this task, where continuous motion trajectories of each pixel over the target time span are evaluated, and these intermediate states are included in the metrics, our solution significantly improves results. Conversely, for tasks less sensitive to intermediate states and focused only on final outcomes, such as **object detection** or **optical flow estimation**, Labits may not outperform some well-established representations and solutions. In these cases, the additional fine-grained intermediate information may even become irrelevant noise that cannot be properly utilized. Different tasks require different types of information, and while our solution is not a one-size-fits-all approach, it can exhibit substantial advantages for suitable tasks.
> 2. **We did not conduct extensive adaptation experiments across various event camera-based vision tasks.**
>    This paper focuses on providing an exceptional solution for the specific task of dense continuous-time trajectory estimation, rather than proposing a universal representation. Applying Labits is not as simple as directly replacing previously used representations in various tasks. Instead, **it requires tailored feature fusion designs for Labits, APLOF, and a suitable task that focuses on intermediate states**. Therefore, we have to redesign the pipeline case-by-case for different tasks. Redesigning and conducting training, testing, and ablation studies for diverse tasks is beyond the scope of a single conference paper and was not our intention. This work remains centered on the specific task highlighted in the title.
> 3. **Labits omits event density information.**
>    Labits is designed to maximize the extraction and retention of motion information at intermediate time points. However, this design sacrifices event density information, which might be crucial for certain computer vision tasks. Future work could explore combining multiple representations to create a more generalizable framework that balances these trade-offs.
>
> ### Future Directions
>
> This paper introduces Labits along with a series of models and structures that fully exploit its capabilities. We have demonstrated its unparalleled suitability for dense continuous-time trajectory estimation and validated the effectiveness of Labits and its associated structures through comprehensive ablation studies. Potential future research directions include:
>
> 1. Adapting Labits and its associated models to other similar tasks, such as human keypoint tracking, object trajectory tracking, or event-based video interpolation. These adaptations would undoubtedly require redesigned model structures tailored to each task’s specific needs.
>
> 2. Leveraging the properties of active pixels proposed in this work to generate representations for selective regions rather than globally. This approach could enable more efficient tracking by focusing on active pixel clusters.
>
> 3. Combining Labits with traditional event representations like Voxel Grid to achieve complementary strengths. Labits captures fine-grained temporal information for extracting local motion states at intermediate moments but lacks event density information. Conversely, Voxel Grid provides event density information but discards the fine-grained temporal precision that event cameras excel at. Combining these representations may open new possibilities for many tasks.
>
> 4. Labits, for efficiency and information density considerations, does not differentiate between events of different polarities. Exploring more optimized handling of polarity-specific events may further enhance its performance in relevant computer vision tasks.
>
> In conclusion, the innovations proposed in this paper are versatile and can be extended or improved upon. We hope more researchers will contribute to developing better event representations, unlocking the full potential of event cameras.

---

> ### Author Response · Authors · 2024-11-22
> **Q2, Q3, Q6: Comparison with Existing Representations & Clarification on Voxel Grids and Event Polarity**
>
> ## Q2 & Q6: Comparison with Existing Representations:
>
> **Response**: In this paper, we conducted extensive experiments to validate the effectiveness of Labits in trajectory estimation. As detailed in **Table 2** of our paper, incorporating Labits led to substantial improvements across key trajectory estimation metrics.
>
> Regarding Q2, which mentioned the absence of a direct comparison with DCT-RAFT, we believe there is a small misunderstanding. In **Table 2** of the main paper:
> - **"Ours w/o APLOF Features"** refers to **"DCT-RAFT using Labits (with the same set of modifications as our method)"**.
> - **"Ours w/o APLOF & Labits"** refers to **"DCT-RAFT using Voxel (with the same set of modifications as our method)"**.
>
> Thus, the results provided in **Table 2** in our paper present the requested comparisons. To clarify, we have renamed the method column for better understanding and highlighted the relevant rows below for your reference.
>
> The results clearly demonstrate a significant reduction in all metrics when replacing Voxel with Labits, showcasing the latter's superior performance.
>
> | Method                     | Input | TEPE ↓       | TAE ↓        | EPE ↓        | AE ↓         |
> |----------------------------|-------|--------------|--------------|--------------|--------------|
> | Labits-RAFT (Ours)                           | E       | 1.32     | 3.14     | 2.50     | 3.37     |
> | Labits-RAFT (Ours)                           | E+I     | 0.66     | 1.72     | 1.08     | 1.45     |
> | DCT-RAFT using Labits (Modified as Ours)     | E       | 1.53     | 3.69     | 2.83     | 3.89     |
> | **DCT-RAFT using Labits (Modified as Ours)** | **E+I** | **1.01** | **2.63** | **1.64** | **2.25** |
> | DCT-RAFT using Voxel (Modified as Ours)      | E       | 1.83     | 4.57     | 3.29     | 4.74     |
> | **DCT-RAFT using Voxel (Modified as Ours)**  | **E+I** | **1.16** | **3.01** | **1.99** | **2.78** |
> | DCT-RAFT using Voxel (Original)              | E       | 1.85     | 4.61     | 3.37     | 4.80     |
> | DCT-RAFT using Voxel (Original)              | E+I     | 1.29     | 3.35     | 2.27     | 3.19     |
>
>
> ## Q3: Clarification on Voxel Grids and Event Polarity:
>
> **Response**: Thank you for the inquiry regarding the information in **Table 1**. The table is extended from references [1] and [2].
>
> We interpret the statement "voxel grid discards the event polarity" to mean that, during voxel grid generation, events with different polarities are not calculated separately, as is done in representations like time surfaces, event count, or TORE volume. More specifically, the voxel grid representation has a shape of $B \times H \times W$ rather than $2 \times B \times H \times W$.
>
> Additionally, voxel grids accumulate events of different polarities at each location by treating polarity as signs (±). This approach causes the resulting values to lose explicit polarity information, thus cannot be treated as fully retaining the polarity information.

---

> ### Author Response · Authors · 2024-11-22
> **Q4, Q5, Q7, W1, W2: Further Comparison with Voxel Grid, Clarifications on Equation Parameters and Calculations**
>
> ## Q4: Voxel-to-APLOF Network Training:
>
> **Response**: Thanks for recommending the comparative experiments involving the training of a Voxel-to-APLOF network and its integration into the RAFT model pipeline. We conducted these experiments under uniform training conditions, specifically training the Voxel-to-APLOF network. For training the Voxel-to-APLOF network, we employ a zero mask to get active pixels. The **table (Table 7 & Table 8 in the paper)** below provides a brief comparison of the minimal total loss (**Equation [7]**) between the Voxel-to-APLOF network and the Labits-to-APLOF network.
>
> | Model                | Total Loss | LR APLOF L1 Loss | HR APLOF L1 Loss |
> |----------------------|------------|------------------|------------------|
> | Labits-to-APLOF Net  | 0.084      | 0.056            | 0.028            |
> | **Voxel-to-APLOF Net**   | **0.217**      | **0.080**            | **0.137**            |
>
> This result is expected since the Labits/Voxel-to-APLOF net only takes in one time bin's representation layer per inference, and voxel grid layers by theory doesn't contain any time gradient-like information to conduct active pixel local optical flow estimation.
>
> Furthermore, we integrated the pretrained Voxel-to-APLOF network into the RAFT model pipeline for trajectory estimation. By maintaining all conditions consistent with our Labits-RAFT model—except for the use of voxel and Voxel-to-APLOF features and removing the APM—we generated the results shown in the subsequent **table**.
>
> | Method                     | Input | TEPE         | TAE          | EPE          | AE           |
> |----------------------------|-------|--------------|--------------|--------------|--------------|
> | DCT-RAFT                   | E+I   | 1.29         | 3.35         | 2.27         | 3.19         |
> | Labits-RAFT (Ours)         | E+I   | 0.66         | 1.72         | 1.08         | 1.45         |
> | **Voxel-to-APLOF Net + DCT-RAFT**  | **E+I**   |    **0.84**      |    **2.25**      |  **1.40**        |   **1.97**       |
>
> The results highlight Labits' clear advantage in instantaneous active pixel local optical flow estimation compared to the voxel grid, further validating the effectiveness of both Labits and the Labits-to-APLOF Net.
>
>
> ## Q5: Clarifications on Equation Parameters and Calculations:
>
> **Response**: Sorry for the misleading description. $\tau_{start}$ and $\tau_{end}$ represent the first and last event timestamps of a given event stream, and a pixel x's coordinate at $\tau_{\text{start}}$ are denoted as $\mathbf{x_{\text{start}}}$. All optical flows mentioned in equations (5) and (6) refer to the optical flow (or pixel-level accumulated displacement) between the start time and the time indicated in the subscript. For example, $O_{\tau}(\mathbf{x})$ denotes the optical flow between the start time and the specified time $\tau$. Lastly, $O_{\tau}, O_{\tau+}, O_{\tau-}$ are ground truth values provided in the dataset and are not derived from our calculations.
>
> ## Q7 & W1 & W2: Experimentation on Additional Datasets & Other Tasks:
>
> **Response**: We appreciate the suggestions. However, as detailed in our response to Q1, the full potential of Labits can only be realized in tasks that emphasize intermediate states and datasets that provide ground truth for such states. While DSEC is an optical flow estimation dataset and may appear similar, it is fundamentally different from the dense trajectory estimation task. Optical flow estimation focuses solely on the accumulative displacement of pixels during the target time period, whereas dense trajectory estimation targets the entire intermediate trajectory.
>
> DSEC does not offer intermediate guidance or ground truth, which makes it impossible to properly train and evaluate the fine-grained temporal gradients provided by Labits. In such tasks, the fine-grained temporal information, a key strength of Labits, may even be treated as noise. Therefore, conducting appropriate experiments on this dataset is not feasible.
>
> Similarly, as described in our response to Q1, while Labits could theoretically benefit other tasks like human keypoint trajectory estimation or object trajectory estimation, these would require significant effort to reconstruct corresponding model structures and conduct separate ablation studies. Such extensive adaptations are beyond the scope of this work.
>
> This paper focuses on presenting a dedicated solution for event-based dense continuous-time trajectory estimation. Notably, we achieved approximately a 50% improvement across all metrics, which we consider a significant contribution. We hope that transferring the proposed method to other related tasks is not deemed essential for the acceptance of this work.

---

> ### Author Response · Authors · 2024-11-22
> **W3, W4, References: Visualization of Time-Continuous Prediction Results & Detailed Ablation Studies and Parameter Analyses**
>
> ## W3: Visualization of Time-Continuous Prediction Results:
>
> **Response**: Thank you for your suggestion regarding the presentation of dense, time-continuous prediction results. We have incorporated visualizations of these outcomes in **Figure 9** of our paper, which includes ten intermediate instances of OF, HR APLOF, and LR APLOF within a sample window from the MultiFlow dataset. Each visualization is aligned with the timestamps of the ground truth to provide a clear, comparative perspective. The trajectory spans a time range of 0.5 seconds (context). We evenly sampled 10 intermediate timestamps, and show the predicted and groud truth optical flows between them and the start time. Meanwhile, APLOFs show instantaneous optical flows.
>
>
> ## W4: Detailed Ablation Studies and Parameter Analyses:
> **Response**: We acknowledge and value the highlighted need for ablation on time bin sizes. To facilitate comprehensive evaluations, we have conducted detailed ablation studies focused on different bin configurations for Labits and Voxel Grid in the context of trajectory estimation. Our study design covered a variety of scenarios by setting bin time spans at 0.0125s, 0.0250s, 0.0500s, and 0.1000s for better comparison with DCT-RAFT[3]. The following **table (or Table 6 in the paper)** provides robust results for thoroughly validating the efficacy of Labits and for an effective comparison of its performance against the voxel grid in trajectory estimation.
>
> | Model                 |  Bin Number | Bin Size | TEPE          | TAE           | EPE           | AE            |
> |-----------------------|-----------|-----------|---------------|---------------|---------------|---------------|
> | DCF-RAFT (voxel)      |    9 | 0.1000    |1.81          | 4.53          | 2.87          | 3.97          |
> | **Labits-RAFT (Labits)** |  **9** | **0.1000**         | **0.88 (↓ 51%)**  | **2.35 (↓ 48%)**  | **1.43 (↓ 50%)**  | **1.97 (↓ 50%)**  |
> | DCF-RAFT (voxel)      |   17 | 0.0500    | 1.51          | 3.82          | 2.61          | 3.63          |
> | **Labits-RAFT (Labits)** |  **17**     |**0.0500**  | **0.75 (↓ 50%)**  | **1.94 (↓ 49%)**  | **1.22 (↓ 53%)**  | **1.66 (↓ 54%)**  |
> | DCF-RAFT (voxel)      |    33 | 0.0250   |    1.40          | 3.56          | 2.45          | 3.41          |
> | **Labits-RAFT (Labits)** |    **33**    | **0.0250** | **0.72 (↓ 49%)**  | **1.86 (↓ 48%)**  | **1.17 (↓ 52%)**  | **1.58 (↓ 54%)**  |
> | DCF-RAFT (voxel)      |       65   |0.0125 | 1.29          | 3.35          | 2.27          | 3.19          |
> | **Labits-RAFT (Labits)** |  **65**   |**0.0125** |  **0.66 (↓ 49%)** | **1.72 (↓ 49%)** | **1.08 (↓ 52%)** | **1.45 (↓ 55%)** |
>
> In the **table** above, the metrics decreases noted in parentheses indicate the percentage reduction in error compared to the voxel grid representation under the same time bin number setting. These results highlight three key observations:
>
> 1. **Superior Performance**: Labits consistently outperforms the voxel grid representation across all time bin settings, demonstrating its effectiveness and superior performance in trajectory estimation tasks.
>
> 2. **Impact of Temporal Granularity**: As the number of bins increases, the performance of both Labits and the voxel grid improves. However, Labits consistently achieves better results, underscoring the importance of fine-grained temporal information in trajectory estimation.
>
> 3. **Generalizability**: The same Labits-to-APLOF network functions effectively across various bin size scenarios. Moreover, the rate of metric change aligns closely with that of DCT-Raft as the bin size varies, showcasing the generalizability of our pretrained Labits-to-APLOF network.
>
>
> ## References
>
> [1] Baldwin, R. W., Liu, R., Almatrafi, M., Asari, V., & Hirakawa, K. (2022). Time-ordered recent event (tore) volumes for event cameras. IEEE Transactions on Pattern Analysis and Machine Intelligence, 45(2), 2519-2532.
>
> [2] Gehrig, D., Loquercio, A., Derpanis, K. G., & Scaramuzza, D. (2019). End-to-end learning of representations for asynchronous event-based data. In Proceedings of the IEEE/CVF International Conference on Computer Vision (pp. 5633-5643).
>
> [3] Gehrig, M., Muglikar, M., & Scaramuzza, D. (2024). Dense continuous-time optical flow from event cameras. IEEE Transactions on Pattern Analysis and Machine Intelligence.

---

> ### Author Response · Authors · 2024-11-22
> **Enhanced Paper Content and Supplementary Materials Available**
>
> We have updated the paper content with additional details now available in the supplementary materials. This includes more ablation and comparison results, as well as supportive figures and tables. Please feel free to review the supplementary materials at your convenience, depending on your need. Thanks again!
>
> **[Final Note] Thanks again for the insightful review. If there’s anything else we can clarify or elaborate on, please don’t hesitate to let us know. If our responses have addressed your concerns, we would be grateful for your support in improving our score.**

---

> ### Comment · Reviewer_nUyE · 2024-11-23
> **Event representations**
>
> Thank you for the added analyses. Would you consider more recent event representations to enrich the comparisons and verify the superiority of your proposed method?
>
> E.g.,
> [Learnable MLP] "End-to-end learning of representations for asynchronous event-based data." ICCV, 2019.
>
> [Recurrent Surface] "A differentiable recurrent surface for asynchronous event-based data." ECCV, 2020.
>
> [Former-Latter Event Groups] "Spike-flownet: event-based optical flow estimation with energy-efficient hybrid neural networks." ECCV, 2020.
>
> [Gaussian Weighted Polarities] "Spatio-temporal recurrent networks for event-based optical flow estimation." AAAI, 2022.
>
> [EvSurface] "Learning from images: A distillation learning framework for event cameras." TIP, 2021.
>
> [EvRepSL] "EvRepSL: Event-Stream Representation via Self-Supervised Learning for Event-Based Vision." TIP, 2024.

---

> > ### Author Response · Authors · 2024-11-26
> > **Response to Reviewer's Comments on Comparison with Additional Event Representations**
> >
> > **Optimized and Condensed Response:**
> >
> > We sincerely thank the reviewer for their insightful suggestions regarding the inclusion of recent event representations in our comparisons. We have thoroughly evaluated the feasibility of each recommended method for dense trajectory estimation:
> >
> > 1. **Learnable MLP and Recurrent Surface**: Adapting these representations in our RAFT model is impractical due to their excessive memory and computational demands. For instance, handling 65 time bins at a resolution of 384×512 on the MultiFlow dataset caused out-of-memory errors even with a batch size of 1 on a 24GB GPU. In contrast, their original use case, the N-Caltech101 dataset, operates at a much lower resolution (180×240). Thus, these methods are unsuitable for our task.
> >
> > 2. **Former-Latter Event Groups and Gaussian Weighted Polarities**: These approaches were implemented and tested, aligning well with the layered representation requirements and fitting within our computational constraints.
> >
> > 3. **EvSurface and EvRepSL**: While innovative, these methods lack the dense temporal representation required by RAFT-based models. Specifically, they do not partition the event stream into time bins (shape: B×H×W), making them incompatible with our framework.
> >
> > We prioritized representations that were computationally feasible and ensured fair comparisons for our task. While time and resource constraints prevented us from completing full training at this point, we provide preliminary results at epochs 30, 40, and 50 (of 100 total epochs), which consistently demonstrate the superiority of our method. Should final results become available before the discussion deadline, we will update this response accordingly.
> >
> > **Result Table for 30 epochs of training**
> >
> > | Method                     | Input | TEPE ↓       | TAE ↓        | EPE ↓        | AE ↓         |
> > |----------------------------|-------|--------------|--------------|--------------|--------------|
> > | Former-Latter Event Groups                | E+I     | 4.34     | 10.51     | 5.80     | 7.82     |
> > | Gaussian Weighted Polarities                                            | E+I     | 4.59     | 10.78     | 6.16     | 8.23     |
> > | DCT-RAFT                                                                | E+I     | 2.16     | 5.65     | 3.70     | 5.37     |
> > | **Labits-RAFT (w/o APLOF features)**                                    | E+I     | **1.52**     | **4.13**     | **2.43**     | **3.33**     |
> > | **Labits-RAFT**                                                         | E+I     | **1.36**     | **3.69**     | **2.29**     | **3.04**     |
> >
> > **Result Table for 40 epochs of training**
> >
> > | Method                     | Input | TEPE ↓       | TAE ↓        | EPE ↓        | AE ↓         |
> > |----------------------------|-------|--------------|--------------|--------------|--------------|
> > | Former-Latter Event Groups          | E+I     | 4.22     | 9.84      | 5.54     | 7.33     |
> > | Gaussian Weighted Polarities                                       | E+I     | 4.38     | 10.36     | 5.76     | 7.72     |
> > | DCT-RAFT                                                           | E+I     | 1.94     | 5.24      | 3.44     | 5.28     |
> > | **Labits-RAFT (w/o APLOF features)**                               | E+I     | **1.51**     | **4.05**     | **2.41**     | **3.29**     |
> > | **Labits-RAFT**                                                    | E+I     | **1.13**     | **3.11**     | **1.82**     | **2.67**     |
> >
> > **Result Table for 50 epochs of training**
> >
> > | Method                     | Input | TEPE ↓       | TAE ↓        | EPE ↓        | AE ↓         |
> > |----------------------------|-------|--------------|--------------|--------------|--------------|
> > | Former-Latter Event Groups                           | E+I     | 4.01     | 9.39      | 5.17     | 6.99     |
> > | Gaussian Weighted Polarities                                                       | E+I     | 4.39     | 10.24     | 5.77     | 7.54     |
> > | DCT-RAFT                                                                           | E+I     | 1.68     | 4.46      | 2.90     | 4.15     |
> > | **Labits-RAFT (w/o APLOF features)**                                               | E+I     | **1.32**     | **3.55**     | **2.14**     | **3.04**     |
> > | **Labits-RAFT**                                                                    | E+I     | **0.93**     | **2.57**     | **1.49**     | **2.10**     |
> >
> > We hope this clarifies our selection process and the challenges we faced in addressing the comparisons as suggested. We believe our methodology remains robust and competitive within the scope of feasible comparisons.
> >
> > If you think all our additional results and clarification are sufficient, **we would be extremely grateful for your support in improving our score**. Thanks!

---

> > > ### Comment · Reviewer_nUyE · 2024-11-26
> > > **Comment**
> > >
> > > Thank you for adding the results for comparison against recent event representations, which better verify the superiority and robustness of the proposed method. The results and analyses should be incorporated into the final version. We would like to elevate the rating.

---

> > > > ### Author Response · Authors · 2024-11-26
> > > > **Comment**
> > > >
> > > > Thank you for your feedback and the improved rating! We will include the comparison analysis in our final paper as soon as the experiments are completed. We anticipate submitting the revised manuscript soon.

---

> > > > > ### Comment · Reviewer_nUyE · 2024-11-26
> > > > > **Experiments on real data**
> > > > >
> > > > > The reviewer would still strongly suggest that the authors add some results on real-world datasets like DSEC-Flow. Even if it does not provide high-FPS ground truth, some qualitative results would be nice for enriching the analyses and helping understand the effectiveness of the proposed method. Some metrics like FWL which do not require ground truth could also be used.

---

> > > > > > ### Author Response · Authors · 2024-11-28
> > > > > > **response**
> > > > > >
> > > > > > Dear Reviewer nUyE,
> > > > > >
> > > > > > Thank you for your valuable suggestions. We are committed to conducting further analysis and will provide updates within the allowed timeframe.
> > > > > >
> > > > > > Additionally, we have included a new table (**Table 9**) with the results of more recent event representations in our paper. We sincerely appreciate your insightful feedback.
> > > > > >
> > > > > > Authors.

---

### Official Review · Reviewer_8m1Q · 2024-10-25

**Soundness:** 2
**Presentation:** 3
**Contribution:** 3
**Rating:** 6
**Confidence:** 2

**Summary:**

They introduced Labits, a novel event representation that simultaneously retains fine-grained temporal information, meaningful 2D visual patterns, and local speed cues. They also developed the Labits-to-APLOF net, which accuratelyconverts Labits into active pixel local optical flows. The results high-light that, beyond model architecture, event representations also have a transformative impact onthe final outcomes.

**Strengths:**

1. a novel synchronous event representation that is aware of event camera's asynchronous characteristic and keeps rich local movement trend information

2.They achieved the SOTA performance on event-based dense trajectory estimation task.

3.trained a corresponding active pixel local optical flow estimator based on Labits layers, which significantly benefits movement-sensitive tasks.

**Weaknesses:**

1.In Figure 4, there is less difference between (d) and (e). What’s the influence of APLOF? Could you give explanation for this phenomenon?

2.In Table3, you already give the model size and other metrics. Could you please compare your model with others in FLOPs and total parameters to demonstrate your model’s efficiency?

**Questions:**

See weakness above.

---

> ### Author Response · Authors · 2024-11-22
> **Answering Questions & Addressing Weaknesses**
>
> Dear Reviewer 8m1Q,
>
> We sincerely appreciate the time and effort you have invested in reviewing our paper. We are also thankful for your acknowledgment of the practical significance and contributions of our research.
>
> In response to your comments and suggestions, we will offer clarifications and further details to address the issues highlighted, ensuring that the merits and implications of our work are better understood.
>
> ## W1: Explanation for the Influence of APLOF
>
> **Response**: Thank you for the feedback regarding the clarity of the influence APLOF features have.
>
> Firstly, visualization (e) showcases three trajectory estimation results produced by our complete pipeline, while visualization (d) presents the results obtained from our model without the APLOF features. We acknowledge that the figure contains some ambiguities, and we have updated **Figure 4** in our paper to better emphasize the differences in trajectory estimation outcomes.
>
> Furthermore, the data in **Table 2** of our paper (and in the following **table**) clearly demonstrates the effectiveness of APLOF features from a quantitative perspective. These results provide solid evidence of the significant contribution APLOF features make to trajectory estimation performance.
>
> | Method                       | Input | TEPE          | TAE           | EPE          | AE           |
> |------------------------------|-------|---------------|---------------|---------------|---------------|
> | Ours w/o APLOF Features      | E     | 1.53          | 3.69          | 2.83         | 3.89          |
> | Ours w/o APLOF Features      | E+I   | 1.01          | 2.63          | 1.64         | 2.25          |
> | **Labits-RAFT (Ours)**       | **E**   | **1.32 ↓ 13.7%** | **3.14 ↓ 14.9%** | **2.50 ↓ 11.7%** | **3.37 ↓ 13.4%** |
> | **Labits-RAFT (Ours)**       | **E+I** | **0.66 ↓ 34.7%** | **1.72 ↓ 34.6%** | **1.08 ↓ 34.2%** | **1.45 ↓ 35.6%** |
>
> ## W2: Comparison of model
>
> **Response**: Thank you for highlighting concerns about computational efficiency. As detailed in W1, Labits can enhance trajectory estimation performance without introducing more parameters to the model pipeline. We've included a robust efficiency comparison between DCT-RAFT and Labits-RAFT in the **table (or Table 3 in the paper)** below, based on 100 iterations on an NVIDIA-A10 GPU. Despite the increase in parameters, the inference time remains reasonably contained.
>
>
> | Description           | Trainable Parameters | Non-trainable Parameters | Total Parameters | Estimated Model Size (MB) | Average GFLOPS  | Average Inference Time (s) | TEPE | TAE
> |-----------------------|----------------------|--------------------------|------------------|---------------------------|---------|--------------------|---------|---------|
> | DCT-RAFT   | 6.8 M                | 0                        | 6.8 M            | 27.203                    | 4316.287 | 0.133              | 1.29         | 3.35
> | Labits-RAFT (w/o APLOF features)| 6.8 M               | 0                     | 6.8 M           | 27.203                   | 4432.353 | 0.135              | **1.01 ↓ 22%**   | **2.63 ↓ 21%**
> | Labits-RAFT (with APLOF features)| 24.8 M               | 524 K                    | 25.3 M           | 101.288                   | 5599.052 | 0.192              | **0.66 ↓ 49%**   | **1.72 ↓ 49%**
>
> Additionally, we already provided the time consumption comparison for various existing event representations. Labits was evaluated for efficiency using 500 event packets from the MultiFlow validation set, achieving an average execution time of 0.220 seconds. Details can be found in subsection **Efficiency Analysis in section 5.Results**.
>
> We have updated the paper content with additional details now available in the supplementary materials. This includes more ablation and comparison results, as well as supportive figures and tables. Please feel free to review the supplementary materials at your convenience, depending on your need. Thanks again!
>
> **[Final Note] Thanks again for the insightful review. If there’s anything else we can clarify or elaborate on, please don’t hesitate to let us know. If our responses have addressed your concerns, we would be grateful for your support in improving our score.**

---

> > ### Comment · Reviewer_8m1Q · 2024-11-23
> > **Response**
> >
> > Thank you for your response. I have no concerns.

---

> > > ### Author Response · Authors · 2024-12-03
> > > **Kindly Request for a Score Raise**
> > >
> > > Dear Reviewer 8m1Q,
> > >
> > > Thank you for your kind words and appreciation of our work. In your previous comments, you mentioned that you did not have concerns about this work. During the rebuttal session, we significantly improved the paper by adding more plots, experiments, and ablations. The latest version of our paper PDF includes these additional contents, which further support the superiority of our proposed method.
> > >
> > > As the current average score of our paper is borderline, we kindly request a potential score raise if you find our work worthy. **Your support and recognition would be immensely valuable for the acceptance of this paper.**
> > >
> > > We are **extremely grateful** for your consideration and kindness.

---

### Official Review · Reviewer_zWL1 · 2024-11-01

**Soundness:** 3
**Presentation:** 3
**Contribution:** 3
**Rating:** 6
**Confidence:** 3

**Summary:**

This paper introduces Labits, a new representation method called "Layered Bidirectional Time Surfaces," designed for event cameras to enable precise dense trajectory estimation and optical flow calculation. Labits maximize the advantages of event cameras by capturing fine-grained temporal information, retaining stable 2D features, and maintaining consistent information density in spatiotemporal motion estimation. The authors also propose a specialized module, Labits-to-APLOF (Active Pixel Local Optical Flow), which uses Labits representations to enhance the accuracy of local optical flow estimation at active pixels. This paper demonstrates Labits' superior performance by reducing trajectory end-point error (TEPE) by 49% over previous state-of-the-art methods in the MultiFlow dataset, making it an effective and efficient tool for motion-sensitive tasks.

**Strengths:**

- The motivation is compelling, addressing the need for more accurate and dense event representations that fully exploit the unique properties of event cameras. This need is crucial in dynamic environments where event cameras can provide high temporal resolution.
- The paper is organized logically, with a clear distinction between the problem background, the proposed solution (Labits), and the experiments.
- Labits and the Labits-to-APLOF module demonstrate notable improvements in trajectory estimation accuracy. By leveraging bidirectional time surfaces, Labits successfully addresses temporal occlusion and provides a stable representation of movement information, which is particularly useful for complex tasks such as trajectory prediction, event-based video interpolation, and high-speed object tracking.

**Weaknesses:**

- Some technical justifications for the Labits representation could be expanded. For example, while the use of bidirectional time surfaces is novel, more in-depth analysis of why this bidirectional approach provides such significant accuracy improvements over unidirectional methods could make the claims stronger.
- The paper introduces the Labits-to-APLOF network as a lightweight and adaptable feature. However, additional evidence supporting its adaptability, such as visual examples of Labits in various scenarios, would clarify its generalizability.
- While Labits achieves efficiency gains, it is slower than simpler methods like the Voxel Grid. Additional experiments demonstrating the trade-off between computational cost and accuracy gains in real-time settings would strengthen the justification for using Labits in time-sensitive applications.
- The evaluation of Labits is confined to a single dataset (MultiFlow), limiting insight into its performance across different event-based vision tasks and datasets. Exploring Labits in various contexts and testing it on multiple datasets would offer a more comprehensive assessment of its versatility and robustness, providing a promising direction for future research.

**Questions:**

- The authors claim that the bidirectional time surface design enables Labits to reduce estimation error significantly. Could they provide additional analysis or visualizations to show why this bidirectional approach better captures motion density and consistency compared to unidirectional methods?
- Could the authors clarify the Labits layer's adaptability? Specifically, how does Labits handle scenarios with minimal event data where forward and backward time data might be sparse?

---

> ### Author Response · Authors · 2024-11-22
> **Answering Questions & Addressing Weaknesses**
>
> Dear Reviewer zWL1,
>
> We sincerely appreciate the time and effort you have dedicated to reviewing our paper and recognizing the practical contributions of our research.
>
> In response to your comments and suggestions, we will provide clarifications and additional details to address the concerns raised, ensuring a clearer understanding of the merits and implications of our work.
>
> ## Q1 & W1: Additional Analysis & Visualization of Bidirectional Structure
>
> **Response**: Thanks for proposing this valuable question. Indeed, since **bidirectional** is a key factor of our event representation, we ought to provide more concrete evidence to show the effectiveness of this design scheme. We conducted additional ablation studies on whether the representation takes in **bidirectional** information during the representation generation. For the so-called **One-way Labits**, only events triggered during the past time range is considered, following the traditional time surface generation strategy. We retrained the entire pipeline using the **One-way Labits**, including the Labits-to-APLOF net and the Labits-RAFT model. The results of both model is significantly worse compared to our proposed bidirectional Labits. Details are shown in the **following table and additional figures** (**Table 4, Table 5, and Figure 12** in our paper).
>
> The minimal total loss (**Equation [7]**) during the training:
>
> | Model                | Total Loss | LR APLOF L1 Loss | HR APLOF L1 Loss |
> |----------------------|------------|------------------|------------------|
> | Labits-to-APLOF Net  | 0.084      | 0.056            | 0.028            |
> | **Labits-to-APLOF Net (One-way Labits)**   |   **0.357**    |   **0.199**       |    **0.158**      |
>
> The Model performance of DCR-RAFT, Labits-RAFT(Labits), and Labits-RAFT(One-way Labits)
>
> | Method                     | Input | TEPE         | TAE          | EPE          | AE           |
> |----------------------------|-------|--------------|--------------|--------------|--------------|
> | DCT-RAFT                   | E+I   | 1.29         | 3.35         | 2.27         | 3.19         |
> | Labits-RAFT (Ours)         | E+I   | 0.66         | 1.72         | 1.08         | 1.45         |
> | **Labits-RAFT (One-way Labits)**    | **E+I**   |   **0.72**       |   **1.87**       |   **1.22**       |    **1.68**      |
>
> ## Q2: Clarification of Labits Layer's Adaptability
>
> **Response**: When events are very sparse, any event representation will encounter challenges due to the inherently lower information content during the same time period. However, Labits addresses this issue more effectively compared to voxel grids for the following reasons:
>
> 1. **Reduced Dependence on Event Density**: The value in voxel grids is heavily influenced by event density, whereas the value in Labits is determined solely by the relative time difference between the closest event's timestamp and the probe time. This independence makes Labits less sensitive to event sparsity.
> 2. **Bidirectional Time Range Utilization**: Each layer of a voxel grid only considers events from the past, exacerbating sparsity issues by ignoring future events. In contrast, Labits incorporates events from both the past and the future. As long as at least one event occurs within this range at a given pixel, Labits can maintain a meaningful value.
>
> Therefore, while event sparsity remains a objective limitation, Labits is better equipped to mitigate its effects compared to voxel grids.
>
> ## W2: Visual Examples of Labits in Various scenarios
>
> **Response**: Thanks for raising this concern. We have generated Labits on three well-known event camera datasets: MVSEC[3], DSEC[1], and MultiFlow[2]. The time ranges and the number of bins were adjusted to accommodate the data formats of each dataset and to achieve better visual clarity. Detailed visualizations can be found in the **Figure 10** in our paper.

---

> ### Author Response · Authors · 2024-11-22
> **Additional Efficiency Analysis**
>
> ## W3: Additional Efficiency Analysis
>
> **Response**: Thanks for highlighting concerns about computational efficiency. As detailed in W1, Labits enhances trajectory estimation performance without adding more parameters to the model pipeline.
>
> We have provided a comparison of time consumption across various existing event representations. Labits was evaluated for efficiency using 500 event packets from the MultiFlow validation set, achieving an average execution time of 0.220 seconds. Detailed results can be found in subsection **Efficiency Analysis in Section 5: Results**, or in the **table** below.
>
> | Representation | Average Generation Time Cost (s) |
> | -------------- | -------------------------------- |
> | Event Frame    | 0.062                            |
> | Event Count    | 0.102                            |
> | Labits         | 0.220                            |
> | Voxel Grid     | 0.225                            |
> | Time Surface   | 0.358                            |
> | TORE Volume    | 7.624                            |
>
> While Labits generation is not the fastest, it is slightly faster than voxel grid generation and demonstrates fairly efficient performance considering its computational complexity.
>
> We've also included a robust efficiency comparison between DCT-RAFT and Labits-RAFT in the **table (or Table 3 in the paper)** below, based on 100 iterations on an NVIDIA-A10 GPU. Despite the increase in parameters, the inference time remains reasonably contained. The increase in inference time is justified by the significant reductions in error and enhancements in model performance. This trade-off is acceptable within the context of the gains achieved.
>
>
> | Description           | Trainable Parameters | Non-trainable Parameters | Total Parameters | Estimated Model Size (MB) | Average GFLOPS  | Average Inference Time (s) | TEPE | TAE
> |-----------------------|----------------------|--------------------------|------------------|---------------------------|---------|--------------------|---------|---------|
> | DCT-RAFT   | 6.8 M                | 0                        | 6.8 M            | 27.203                    | 4316.287 | 0.133              | 1.29         | 3.35
> | Labits-RAFT (w/o APLOF features)| 6.8 M               | 0                     | 6.8 M           | 27.203                   | 4432.353 | 0.135              | **1.01 ↓ 22%**   | **2.63 ↓ 21%**
> | Labits-RAFT (with APLOF features)| 24.8 M               | 524 K                    | 25.3 M           | 101.288                   | 5599.052 | 0.192              | **0.66 ↓ 49%**   | **1.72 ↓ 49%**

---

> ### Author Response · Authors · 2024-11-22
> **Discussion on Limitations and Future Works**
>
> ## W4: Limitations of Labits
>
> **Response**: Thanks for highlighting the weaknesses of our work. We have identified the limitations in this study and outlined future directions that could effectively address these limitations.
>
> ### Limitations
> 1. **The proposed solution has specific use cases.**
>    Labits demonstrates excellent performance in dense continuous-time trajectory estimation tasks. However, this is not only due to our novel event representation and a well-designed model pipeline but also because the task definition and the selected dataset inherently leverage Labits’ ability to preserve rich intermediate temporal information. In this task, where continuous motion trajectories of each pixel over the target time span are evaluated, and these intermediate states are included in the metrics, our solution significantly improves results. Conversely, for tasks less sensitive to intermediate states and focused only on final outcomes, such as **object detection** or **optical flow estimation**, Labits may not outperform some well-established representations and solutions. In these cases, the additional fine-grained intermediate information may even become irrelevant noise that cannot be properly utilized. Different tasks require different types of information, and while our solution is not a one-size-fits-all approach, it can exhibit substantial advantages for suitable tasks.
>
> 2. **We did not conduct extensive adaptation experiments across various event camera-based vision tasks.**
>    This paper focuses on providing an exceptional solution for the specific task of dense continuous-time trajectory estimation, rather than proposing a universal representation. Applying Labits is not as simple as directly replacing previously used representations in various tasks. Instead, **it requires tailored feature fusion designs for Labits, APLOF, and a suitable task that focuses on intermediate states**. Therefore, we have to redesign the pipeline case-by-case for different tasks. Redesigning and conducting training, testing, and ablation studies for diverse tasks is beyond the scope of a single conference paper and was not our intention. This work remains centered on the specific task highlighted in the title.
>
> 3. **Labits omits event density information.**
>    Labits is designed to maximize the extraction and retention of motion information at intermediate time points. However, this design sacrifices event density information, which might be crucial for certain computer vision tasks. Future work could explore combining multiple representations to create a more generalizable framework that balances these trade-offs.
>
> ### Future Directions
>
> This paper introduces Labits along with a series of models and structures that fully exploit its capabilities. We have demonstrated its unparalleled suitability for dense continuous-time trajectory estimation and validated the effectiveness of Labits and its associated structures through comprehensive ablation studies. Potential future research directions include:
>
> 1. Adapting Labits and its associated models to other similar tasks, such as human keypoint tracking, object trajectory tracking, or event-based video interpolation. These adaptations would undoubtedly require redesigned model structures tailored to each task’s specific needs.
>
> 2. Leveraging the properties of active pixels proposed in this work to generate representations for selective regions rather than globally. This approach could enable more efficient tracking by focusing on active pixel clusters.
>
> 3. Labits, for efficiency and information density considerations, does not differentiate between events of different polarities. Exploring more optimized handling of polarity-specific events may further enhance its performance in relevant computer vision tasks.
>
> In conclusion, the innovations proposed in this paper are versatile and can be extended or improved upon. We hope more researchers will contribute to developing better event representations, unlocking the full potential of event cameras.
>
> ### References
> [1] Gehrig, M., Aarents, W., Gehrig, D., & Scaramuzza, D. (2021). Dsec: A stereo event camera dataset for driving scenarios. IEEE Robotics and Automation Letters, 6(3), 4947-4954.
>
> [2] Gehrig, M., Muglikar, M., & Scaramuzza, D. (2024). Dense continuous-time optical flow from event cameras. IEEE Transactions on Pattern Analysis and Machine Intelligence.
>
> [3] Zhu, A. Z., Thakur, D., Ozaslan, T., Pfrommer, B., Kumar, V., & Daniilidis, K. (2018). The Multi Vehicle Stereo Event Camera Dataset: An Event Camera Dataset for 3D Perception. IEEE Robotics and Automation Letters, 3(3), 2032-2039.

---

> ### Author Response · Authors · 2024-11-22
> **Enhanced Paper Content and Supplementary Materials Available**
>
> We have updated the paper content with additional details now available in the supplementary materials. This includes more ablation and comparison results, as well as supportive figures and tables. Please feel free to review the supplementary materials at your convenience, depending on your need. Thanks again!
>
> **[Final Note] Thanks again for the insightful review. If there’s anything else we can clarify or elaborate on, please don’t hesitate to let us know. If our responses have addressed your concerns, we would be grateful for your support in improving our score.**

---

> > ### Comment · Reviewer_zWL1 · 2024-11-25
> >
> > I appreciate your detailed rebuttal. Most of my concerns have been addressed.
> >
> > However, I agree with Reviewer XbDn on the request for application on real data, which I might have neglected in my previous comments. It would be more convincing if the authors could provide video demos and quantitative results on real data.

---

> ### Author Response · Authors · 2024-12-03
> **Kindly Request for a Score Raise**
>
> Dear Reviewer zWL1,
>
> Thank you for acknowledging that most of your concerns have been resolved during the rebuttal session. Over the past two weeks, we have addressed all potential concerns to the best of our ability, and most have indeed been properly resolved. The only exception is the "real data" experiment mentioned by some reviewers. Unfortunately, the only available relevant event-based dataset, DSEC, is designed for optical flow estimation, which is not suitable for our dense trajectory tracking task simply due to the lack of necessary ground truth. As honest researchers, we cannot claim to adapt a pipeline optimized for task A to seamlessly work for task B, given the distinct focus and significant domain gap. Since this work is not dataset-centric and there is no well-collected "real-world" dataset for this dense trajectory estimation task, we believe this issue is beyond the scope of this paper.
>
> However, during the rebuttal session, we have significantly improved the paper by adding more plots, experiments, and ablations. The latest version of our paper PDF includes these additional contents, where all these additional results further support the superiority of our proposed method.
>
> This paper is well polished with a decent quality. As the current average score of our paper is borderline, we hope it doesn't get rejected due to an issue beyond our control, and kindly request a potential score raise if you find our work worthy. **Your support and recognition would be immensely valuable for the acceptance of this paper.**
>
> We are **extremely grateful** for your consideration and kindness.

---

### Official Review · Reviewer_XbDn · 2024-11-03

**Soundness:** 2
**Presentation:** 2
**Contribution:** 2
**Rating:** 5
**Confidence:** 5

**Summary:**

This paper tackles the representation problem of asynchronous indefinite-length event data feeding into modern networks. The authors propose Labits, a new representation based on a hierarchical bi-directional event surface, to better preserve the millisecond time information in the raw data. The authors conduct experiments on the continuous point tracking task and verify the performance benefits over other representations. However, from my perspective, the contribution is still insufficient, with only a new representation that has not been fully experimentally validated.

**Strengths:**

1. A new event representation Labits is proposed to preserve rich temporal information for motion-related downstream tasks.
2. The benefits of this new representation compared to the commonly used voxel grid are demonstrated on the nonlinear dense trajectory estimation task.

**Weaknesses:**

1. Experimental comparisons only on simulated data are not convincing, especially since the compared methods [1] already did some experiments on real data. At least qualitative or quantitative experimental comparisons should be done on a real collected event data, e.g., the optical flow estimation evaluation on DSEC in [1].

2. The new representations Labits presented in this paper are one of the core contributions, and it is necessary to apply this to multiple tasks (both motion-related and motion-irrelevant) and perform comprehensive evaluations, e.g., two-frame optical flow estimation, depth estimation, object detection, motion segmentation, etc. The authors only experimented with one task and mentioned that it could be used for other motion-related downstream tasks, but not actually provided.

3. ``Since the number of time bins in Labits can be freely adjusted by the user, maintaining a single-channel input increases flexibility.'' Does this indicate that the pretrained Labits-to-APLOF Net can adaptively handle different time bins set by the user? In addition, ablation experiments with a larger number of channels are necessary.

4. The differences between the dense tracking model and [1] need to be clarified.

[1] Dense continuous-time optical flow from event cameras, TPAMI 2024.

**Questions:**

1. Table 1 indicates that the Voxel Grid discards polarity information. How does Labits handle this?
2. ``Labits is a synchronous event representation.'' Does it support online processing?

---

> ### Author Response · Authors · 2024-11-22
> **Q1, Q2: Polarity Information & Online Processing**
>
> Dear Reviewer XbDn,
>
> We sincerely appreciate the time and effort you have invested in reviewing our paper. We value yout feedback and are committed to addressing the concerns you have raised.
>
> We will provide further clarifications and additional information to ensure the significance and contributions of our research are fully understood.
>
> ## Q1: How Labits Handle Polarity Information
>
> **Response**: Labits deliberately discards polarity information for the following reasons:
>
> 1. **Reduced Memory and Computation Costs**: Labits already has a tensor shape of $B \times H \times W$, where $ B $ can be as large as 65 in the MultiFlow dataset. Handling polarity separately would double the tensor size, leading to significant memory consumption and potential out-of-memory (OOM) issues.
> 2. **Increased Information Density and Smoother Time Gradients**: Separating events by polarity results in sparser layers, with more empty pixels on each Labits layer. This sparsity reduces the accuracy of APLOF estimation. By merging polarities, Labits achieves higher information density, allowing smoother and more reliable time gradients.
> 3. **Preservation of Clear Value Semantics**: In Labits, a pixel's value represents the relative time difference between the temporally closest event and the intermediate probe time, with the sign (±) indicating whether the event occurred earlier or later than the probe time. This clear physical meaning is a distinctive feature of Labits. Treating polarity as a sign, as done in some other representations, would obscure this meaning, complicating downstream tasks like APLOF and trajectory estimation.
>
>
> ## Q2: Does Labits Support Online Processing
>
> **Response**: Thank you for bringing up concerns regarding online processing. Although our current Labits generation code is designed for fixed event stream processing, the Labits representation itself theoretically supports online processing. We are actively developing a Labits online processing class and will release it with the codebase once the paper is accepted. This online processing class will enable users to generate Labits representations on-the-fly as new events arrive, facilitating real-time applications and online processing scenarios.

---

> ### Author Response · Authors · 2024-11-22
> **W3, W4: Ablation on Time Bins Number & Clarification of Labits-RAFT**
>
> ## W3: Ablation Expriments with Number of Time Bins
>
> **Response**: Yes, the pretrained Labits-to-APLOF net can adaptively handle different time bins. We acknowledge and value the highlighted need for ablation on time bins using the same pretrained Labits-to-APLOF net. To facilitate comprehensive evaluations, we have conducted detailed ablation studies focused on different bin configurations for Labits (Using the same pretrained Labits-to-APLOF network, which was originally trained with Labits featuring 65 bins and a bin size of 0.0125s) and Voxel Grid, specifically in the context of trajectory estimation. Our study design covered a variety of scenarios by setting bin time spans at 0.0125s, 0.0250s, 0.0500s, and 0.1000s for better comparison with DCT-RAFT[1]. The following **table (or Table 6 in the paper )** provides robust results for thoroughly validating that the pretrained Labits-to-APLOF net can adaptively handle different time bins.
>
> | Model                 |  Bin Number  | Bin Size (s) | TEPE          | TAE           | EPE           | AE            |
> |-----------------------|-----------|-----------|---------------|---------------|---------------|---------------|
> | DCF-RAFT (voxel)      |    9 | 0.1000    |1.81          | 4.53          | 2.87          | 3.97          |
> | **Labits-RAFT (Labits)** |  **9** | **0.1000**         | **0.88 (↓ 51%)**  | **2.35 (↓ 48%)**  | **1.43 (↓ 50%)**  | **1.97 (↓ 50%)**  |
> | DCF-RAFT (voxel)      |   17 | 0.0500    | 1.51          | 3.82          | 2.61          | 3.63          |
> | **Labits-RAFT (Labits)** |  **17**     |**0.0500**  | **0.75 (↓ 50%)**  | **1.94 (↓ 49%)**  | **1.22 (↓ 53%)**  | **1.66 (↓ 54%)**  |
> | DCF-RAFT (voxel)      |    33 | 0.0250   |    1.40          | 3.56          | 2.45          | 3.41          |
> | **Labits-RAFT (Labits)** |    **33**    | **0.0250** | **0.72 (↓ 49%)**  | **1.86 (↓ 48%)**  | **1.17 (↓ 52%)**  | **1.58 (↓ 54%)**  |
> | DCF-RAFT (voxel)      |       65   |0.0125 | 1.29          | 3.35          | 2.27          | 3.19          |
> | **Labits-RAFT (Labits)** |  **65**   |**0.0125** |  **0.66 (↓ 49%)** | **1.72 (↓ 49%)** | **1.08 (↓ 52%)** | **1.45 (↓ 55%)** |
>
> In the **table** above, the metrics decreases noted in parentheses indicate the percentage reduction in error compared to the voxel grid representation under the same time bin number setting. These results highlight three key observations:
>
> 1. **Superior Performance**: Labits consistently outperforms the voxel grid representation across all time bin settings, demonstrating its effectiveness and superior performance in trajectory estimation tasks.
>
> 2. **Impact of Temporal Granularity**: As the number of bins increases, the performance of both Labits and the voxel grid improves. However, Labits consistently achieves better results, underscoring the importance of fine-grained temporal information in trajectory estimation.
>
> 3. **Generalizability**: The same Labits-to-APLOF network functions effectively across various bin size scenarios. Moreover, the rate of metric change aligns closely with that of DCT-Raft as the bin size varies, showcasing the generalizability of our pretrained Labits-to-APLOF network.
>
>
> ## W4: Clarification of Labits-RAFT
>
> **Response**: Thanks for seeking clarification about our model.
> The model in [1] is for the dense optical flow estimation model (referred to as DCT-RAFT hereafter). Our model, Labits-RAFT incorporates several key differences. These distinctions can be summarized as follows:
>
> 1. **Two Independent Stages**: The proposed model is divided into two distinct stages: the *Labits to APLOF Estimation Network* (Stage 1) and the *Labits to Dense Continuous Trajectory Estimation Network* (Stage 2). Stage 1 is a novel contribution of this work and has no equivalent in DCT-Raft.
> 2. **Utilization of Stage 1 Encoder in Stage 2**: The encoder part from Stage 1 is leveraged in Stage 2 to generate APLOF features. These features serve three purposes: (a) enhancing the generation of more informative correlation features, (b) contributing to the creation of context features, and (c) being injected as additional information into the hidden state of the ConvGRU module.
> 3. **Introduction of Active Pixels and Active Pixel Masks (APM)**: Based on events' characteristics, we propose the concept of *Active Pixels* and utilize it to create active pixel masks (APM). Since Labits can only reliably predict accurate speeds in the vicinity of active pixels, APM ensures that accurate information from APLOF features is retained while uncertain information is filtered out.

---

> ### Author Response · Authors · 2024-11-22
> **Discussion on Limitations and Future Works**
>
> ## W1 & W2: Limitations of This Work & Future Research
>
> **Response**: Thank you for pointing out the weaknesses of our work. We have acknowledged the limitations in this study. However, as far as our investigation has shown, some concerns raised regarding these weaknesses, unfortunately, cannot be addressed due to objective reasons. We aim to provide an in-depth discussion of these aspects below. Additionally, we have outlined potential future directions that could effectively address these limitations.
>
> ### Limitations
> 1. **The proposed solution has specific use cases.**
>    Labits demonstrates excellent performance in dense continuous-time trajectory estimation tasks. However, this is not only due to our novel event representation and a well-designed model pipeline but also because the task definition and the selected dataset inherently leverage Labits’ ability to preserve rich intermediate temporal information. In this task, where continuous motion trajectories of each pixel over the target time span are evaluated, and these intermediate states are included in the metrics, our solution significantly improves results. Conversely, for tasks less sensitive to intermediate states and focused only on final outcomes, such as **object detection** or **optical flow estimation**, Labits may not outperform some well-established representations and solutions. In these cases, the additional fine-grained intermediate information may even become irrelevant noise that cannot be properly utilized. Different tasks require different types of information, and while our solution is not a one-size-fits-all approach, it can exhibit substantial advantages for suitable tasks.
> 2. **We did not conduct extensive adaptation experiments across various event camera-based vision tasks.**
>    This paper focuses on providing an exceptional solution for the specific task of dense continuous-time trajectory estimation, rather than proposing a universal representation. Applying Labits is not as simple as directly replacing previously used representations in various tasks. Instead, **it requires tailored feature fusion designs for Labits, APLOF, and a suitable task that focuses on intermediate states**. Therefore, we have to redesign the pipeline case-by-case for different tasks. Redesigning and conducting training, testing, and ablation studies for diverse tasks is beyond the scope of a single conference paper and was not our intention. This work remains centered on the specific task highlighted in the title.
>
> ### Future Directions
>
> This paper introduces Labits along with a series of models and structures that fully exploit its capabilities. We have demonstrated its unparalleled suitability for dense continuous-time trajectory estimation and validated the effectiveness of Labits and its associated structures through comprehensive ablation studies. Potential future research directions include:
>
> 1. Adapting Labits and its associated models to other similar tasks, such as human keypoint tracking, object trajectory tracking, or event-based video interpolation. These adaptations would undoubtedly require redesigned model structures tailored to each task’s specific needs.
>
> 2. Leveraging the properties of active pixels proposed in this work to generate representations for selective regions rather than globally. This approach could enable more efficient tracking by focusing on active pixel clusters.
>
> 3. Combining Labits with traditional event representations like Voxel Grid to achieve complementary strengths. Labits captures fine-grained temporal information for extracting local motion states at intermediate moments but lacks event density information. Conversely, Voxel Grid provides event density information but discards the fine-grained temporal precision that event cameras excel at. Combining these representations may open new possibilities for many tasks.
>
> 4. Labits, for efficiency and information density considerations, does not differentiate between events of different polarities. Exploring more optimized handling of polarity-specific events may further enhance its performance in relevant computer vision tasks.
>
> In conclusion, the innovations proposed in this paper are versatile and can be extended or improved upon. We hope more researchers will contribute to developing better event representations, unlocking the full potential of event cameras.
>
> ## References
> [1] Gehrig, M., Muglikar, M., & Scaramuzza, D. (2024). Dense continuous-time optical flow from event cameras. IEEE Transactions on Pattern Analysis and Machine Intelligence.

---

> ### Author Response · Authors · 2024-11-22
> **Enhanced Paper Content and Supplementary Materials Available**
>
> We have updated the paper content with additional details now available in the supplementary materials. This includes more ablation and comparison results, as well as supportive figures and tables. Please feel free to review the supplementary materials at your convenience, depending on your need. Thanks again!
>
> **[Final Note] Thanks again for the insightful review. If there’s anything else we can clarify or elaborate on, please don’t hesitate to let us know. If our responses have addressed your concerns, we would be grateful for your support in improving our score.**

---

> > ### Comment · Reviewer_XbDn · 2024-11-23
> >
> > Thanks to the authors for their detailed responses, especially the additional experiments. However, I am not completely convinced by the authors' response. I maintain my initial rating at this time.
> >
> > Combining the responses to W1, W2 and W4, the core contribution of this paper is a new event representation, while the network structure applied to the trajectory estimation task is essentially the same as in [1]. The authors do not explicitly specify task-specific relevance of the proposed representation for the trajectory estimation task. Therefore, I argue that the new event representation should be validated on more tasks.
> >
> > The current version only evaluates trajectory estimation on simulated data, which is not convincing. The usability on real data and on different tasks of the new representation can at least be illustrated by a validation on real optical flow datasets according to [1].

---

### Author Response · Authors · 2024-11-22
**General Response**

Dear Reviewers,

We are deeply grateful for your insightful feedback, which has significantly contributed to enhancing our research. Your detailed comments have enabled us to clarify key aspects and substantially strengthen our paper. We have provided separate and detailed responses to each of your comments and suggestions.

In response to your feedback, here are several important updates and clarifications, highlighted in **blue** in the revised draft:

- **Visualizations and Clarifications**: Updated Fig. 4, added Fig. 9 (sample window), Fig. 10 (Labits in various scenarios), and Fig. 12 (One-way Labits).

- **Bidirectional Structure Analysis**: Added Section A.3 in the supplementary material, along with updates to Table 4, Table 5, and Fig. 12.

- **Ablation Study on Time Bin Size**: Detailed in Section A.4 in the supplementary, reflected in Table 6.

- **Additional Comparative Experiments**: Described in Section A.5 of the supplementary, see Table 7 and Table 8.

- **More Efficiency Analysis**: Expanded in Section A.1 of the supplementary, see Table 3.

- **Tone and Overstatement Corrections**: Adjusted in the Abstract, Sections 1, 3, and 6.

- **Limitations and Future Directions**: Discussed in Section A.6 of the supplementary material.

We thank you again for your constructive reviews. We are committed to advancing this field and hope that our revisions meet your expectations and further demonstrate the value of our contributions.

Best regards,

Authors

---

### Author Response · Authors · 2024-11-28
**General Response**

Dear Reviewers,


Thank you for your valuable insights and constructive feedback. We recognize the importance of validating Labits on diverse tasks and real-world datasets, including the two-frame optical flow estimation task with the DSEC dataset. We will work on this and provide detailed comparisons and results in our future updates, as time allows.

We acknowledge that while Labits has specific use cases and prerequisites, it is fundamentally designed to maximize the capabilities of event cameras. Our ambition is for Labits to act as a robust foundational basis that not only fosters further exploration but also encourages the broader research community to leverage its potential in various applications of event-based vision. For a detailed discussion of **the limitations and future directions**, please refer to **Section A.6** of the supplementary material. We trust that this work will inspire new insights and further research within the field.

We are grateful for your guidance and are committed to refining our work based on your insightful feedback.

Additionally, we have included a new table in our paper that compares Labits with more recent event representations (**Table 9**).

Sincerely,

Authors

---

### Meta-Review · Area_Chair_F2qr · 2024-12-24

**Metareview:**

## Summary

The paper introduces Labits, a new representation method for asynchronous indefinite-length event data in modern networks. Labits preserves millisecond time information in raw data by capturing fine-grained temporal information, retaining stable 2D features, and maintaining consistent information density in spatiotemporal motion estimation. The authors also propose a specialized module, Labits-to-APLOF, which enhances local optical flow estimation accuracy at active pixels. Labits reduces trajectory end-point error by 49% over previous methods in the MultiFlow dataset, making it an effective tool for motion-sensitive tasks. Experiments on the MultiFlow dataset demonstrate its effectiveness.

## Strengths

* Presents a novel synchronous event representation, Labits, that preserves rich temporal information for motion-related tasks; demonstrating the benefits of this new representation over the commonly used voxel grid on the nonlinear dense trajectory estimation task.
* The paper is organized logically, distinguishing between the problem background, proposed solution (Labits), and experiments.
* Labits addresses temporal occlusion and provides a stable representation of movement information, useful for complex tasks like trajectory prediction, event-based video interpolation, and high-speed object tracking.
* Achieves SOTA performance on event-based dense trajectory estimation task.

## Weaknesses

* The differences between the dense tracking model and [1] need to be clarified.
* Technical justifications for the Labits representation could be expanded, including the use of bidirectional time surfaces.
* Labits achieves efficiency gains but is slower than simpler methods like the Voxel Grid.
* The evaluation of Labits is confined to a single dataset (MultiFlow), limiting insight into its performance across different event-based vision tasks and datasets. Further experiments could be conducted to show the generalization capacity of the proposed solution.
* More detailed ablation studies and parameter analyses could be conducted.
* The proposed representation should be compared against some existing event representations to verify the superiority of the solution.
* The proposed method achieves high accuracy on MultiFlow, but it is a synthetic dataset.

## Conclusions
Based on the reviews and the author's feedback, the paper should include all the concerns addressed before accepting the paper. Mainly, as stated by three reviewers, they would still strongly suggest that the authors add some results on real-world datasets like DSEC-Flow. Even if it does not provide high-FPS ground truth, some qualitative results would be nice for enriching the analyses and helping understand the effectiveness of the proposed method. Some metrics like FWL which do not require ground truth could also be used.
Moreover, there is a critical concern that were not discussed *Combining the responses to W1, W2 and W4, the core contribution of this paper is a new event representation, while the network structure applied to the trajectory estimation task is essentially the same as in [1]. The authors do not explicitly specify task-specific relevance of the proposed representation for the trajectory estimation task. Therefore, I argue that the new event representation should be validated on more tasks.* and the other two reviewers are also interested on this point.

**Additional Comments On Reviewer Discussion:**

The summary is in the conclusions of the metareview.

---

### Decision · Program_Chairs · 2025-01-22

Reject